# DynamicBias: Sequence-Aware Calibrated Watermarking for Large Language Models

## Abstract

Text watermarking has attracted significant research interest as a way to mitigate LLM-related harms by enabling reliable identification of machine-generated text. In particular, "green" and "red" vocabulary-partition watermarking, which uses a static bias to skew token sampling toward green tokens and away from red, is a promising approach. However, a persistent trade-off remains: stronger watermarks improve detectability but can harm quality, while weaker ones preserve quality but are harder to detect and easier to remove via paraphrasing. A key reason is that the static bias ignores heterogeneous logit distributions across models, domains, and languages, yielding inconsistent performance and hindering practical deployment. Our preliminary investigation shows substantial variability in these distributions and the associated performance disparities, driven by model certainty, measured as the margin between the top logit and the average of a small pool of next-best token logits. Building on this observation, we propose DynamicBias, which calibrates the bias at each step using a sequence-level average of this margin with a single scaling parameter $\alpha$. Theoretically, we show that DynamicBias admits a unique optimal $\alpha$ and increases expected detectability as the sequence-level margin grows. This calibration yields consistent detectability across models and integrates directly with existing vocabulary-partition watermarks, offering a practical solution for real-world deployment. Extensive experiments across four LLMs and three languages demonstrate improved detection with competitive text quality and stronger robustness to paraphrasing. Our code will be available at `https://github.com/[Anonymous]`.

## 1 Introduction

The rapid and pervasive adoption of large language models (LLMs) has unlocked powerful generative capabilities across a wide range of applications (Ouyang et al., 2022; OpenAI et al., 2024; Grattafiori et al., 2024; Qwen et al., 2025). It also raises significant concerns about the traceability and authenticity of AI-generated text, including malicious mass generation of misinformation (Zellers et al., 2019; Chen et al., 2023) and academic dishonesty (Stokel-Walker, 2022; Vasilatos et al., 2025), as well as the risk of future models being inadvertently trained on synthetic content (Radford et al., 2022; Shumailov et al., 2024). Watermarking, which embeds distinctive patterns into generated text, has emerged as a critical approach for provenance and copyright protection (Kirchenbauer et al., 2024; Liu et al., 2024b). However, a persistent trade-off hinders practical deployment: stronger watermarks raise detectability but can harm quality, whereas weaker ones preserve quality but reduce detectability and are easier to paraphrase away (Liu et al., 2024b; Giboulot & Furon, 2024; Wu et al., 2024). Therefore, practical deployment demands watermarks that balance quality and detectability.

Decoding-based watermarking embeds identifiable patterns directly in generated text and has been widely studied. A standard approach is KGW, which partitions the vocabulary into "green" and "red" sets via a prefix hash and adds a static positive bias $\delta$ to "green" logits during generation to encourage their selection (Kirchenbauer et al., 2024). Despite its simplicity, KGW-based approaches often struggle to balance text quality and detectability (Guo et al., 2024; Ren et al., 2024; Chen et al., 2024; Zhao et al., 2024; Wu & Chandrasekaran, 2024). This instability arises because a static bias ignores heterogeneity in logit distributions across models, domains, and languages. While unbiased watermarking aims to preserve the expected sampling distribution to maintain quality (Kuditipudi

et al., 2024; Hu et al., 2024; Wu et al., 2024), current implementations often sacrifice robustness or incur nontrivial computational overhead. Low-entropy watermarking restricts watermarking to uncertain contexts and skips confident tokens (Lu et al., 2024; Liu & Bu, 2024), which helps quality, yet requires the original model at detection time and adds cost. Other works train auxiliary models to improve the trade-off, but these methods are not model-independent and can disrupt end-to-end inference, hindering deployment (Liu et al., 2024a; He et al., 2024; Huo et al., 2024).

Entropy-gated schemes such as SWEET (Lee et al., 2024) gate a static bias $\delta$ by entropy, deciding when to apply the watermark. By contrast, dynamic schemes such as MorphMark (Wang et al., 2025) scale the watermark strength (Wouters, 2024; Takezawa et al., 2025) by the cumulative green-list probability mass. However, both operate at the token-level and do not explicitly capture sequence-level variability, making it difficult to balance text quality and detectability across models and domains. Hence, a practical watermark should be adaptable across models and domains. We address this by calibrating the bias with a sequence-aware signal computed from the logits.

In this paper, we first conduct a preliminary study and find substantial variability in next-token logit distributions across models and domains, with corresponding variability in detectability. These differences reflect model certainty, measured as the gap between the top logit and the mean of the second to fifth largest logits. To address this limitation, we propose DynamicBias, a controller designed to work across models, going beyond token-level heuristics and high-entropy-only targeting. At each step, it measures this gap, maintains a sequence-level average, and calibrates the bias with a single scaling parameter $\alpha$. By aggregating certainty at the sequence level, DynamicBias avoids the under-marking that arises when watermarking is applied only in uncertain regions, thereby preserving quality while improving detectability. Under standard smoothness assumptions, calibrating the bias by the sequence-level margin increases the expected detection statistic in the small-bias regime. DynamicBias integrates directly with vocabulary-partition watermarks.

We conduct experiments and perform an in-depth analysis to demonstrate our proposed method. We use instruction-tuned LLMs from two families: Llama (Grattafiori et al., 2024) and Qwen (Qwen et al., 2025; Yang et al., 2025). We evaluate open-ended generation on C4 (Dodge et al., 2021) and mC4 (Xue et al., 2021), and arithmetic reasoning on GSM8K (Cobbe et al., 2021) and MGSM (Shi et al., 2022) in English, Japanese, and Korean. Experimental results show that integrating DynamicBias into KGW-based methods achieves a better balance between text quality and detectability. Furthermore, DynamicBias remains robust to various paraphrasing attacks, maintaining higher detectability.

Our contributions are summarized as follows: (1) We present preliminary empirical evidence that logit distributions vary substantially across models, domains, and languages, and that this heterogeneity leads to performance disparities for static bias watermarking. (2) We introduce DynamicBias, a sequence-aware mechanism that adapts the bias from a sequence-level average of the logit margin during generation, rather than relying on instantaneous, token-level entropy. (3) We expose a single scaling parameter $\alpha$ that controls bias strength, replacing the ambiguity of a static bias and working across different model families and domains. (4) We demonstrate flexible integration of DynamicBias with existing vocabulary-partition watermarks without modifying hashing, partitioning, or detection.

## 2 PRELIMINARY

The watermarking process for LLMs has two stages: watermark encoding and detection.

**Encoding.** Given a prompt $\mathbf{x}_{1:n} = \{x_1, \ldots, x_n\}$, an LLM $\mathcal{M}$ generates a continuation $\mathbf{x}_{n+1:m} = \{x_{n+1}, \ldots, x_m\}$. We formalize decoding-based watermarking below, using KGW as a representative example (Kirchenbauer et al., 2024), which proceeds as follows.

(1) Compute a hash of the current prefix (e.g., using the last token index as input): $h_{n+1} = H(\mathbf{x}_{1:n})$.

(2) Using $h_{n+1}$, deterministically partition the vocabulary $\mathcal{V}$ into disjoint "green" ($\mathcal{V}_g$) and "red" token sets ($\mathcal{V}_r$).

(3) Let $\ell^{n+1} \in \mathbb{R}^{|\mathcal{V}|}$ be the logits for the next token. Add a static bias $\delta$ ($\delta > 0$) to green logits:

$$\tilde{\ell}_i^{n+1} = \ell_i^{n+1} + \delta \mathbf{1}\{v_i \in \mathcal{V}_g\}. \tag{1}$$

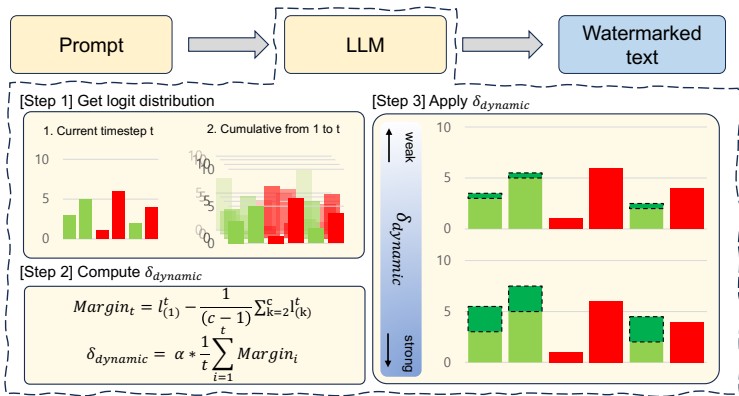

Figure 1: DynamicBias and its sequence-aware calibration.

where $i$ denotes the token index in the vocabulary. Consequently, the bias increases the fraction of "green" tokens in watermarked text relative to typical human text.

**Detection.** An auditor who knows $H$ and the partitioning rule recomputes steps (1) and (2) on the observed sequence and evaluates a $z$-score statistic over the realized count of "green" tokens. Let $T = m - n$ be the number of generated tokens and $G = \sum_{t=n+1}^{m} \mathbf{1}\{x_t \in \mathcal{V}_g\}$ with $\gamma$ denoting the expected fraction of "green" tokens under the null (typically $|\mathcal{V}_g|/|\mathcal{V}|$). The test statistic is

$$z = (G - \gamma T)/\sqrt{T\gamma(1 - \gamma)}. \tag{2}$$

The presence of a watermark is determined by comparing the $z$-score to a pre-defined threshold to test the null hypothesis $H_0$: *The generated text does not follow the green-list constraint*. If $z$ exceeds the threshold, $H_0$ is rejected and the watermark is identified.

**Motivation and KGW-based framework.** Numerous approaches follow the KGW-based (Kirchenbauer et al., 2024) "green" and "red" list design, in which a static bias $\delta$ is applied to green-token logits at each step. However, the static bias mechanism remains unchanged (Guo et al., 2024; Ren et al., 2024; Chen et al., 2024; Zhao et al., 2024; Wu & Chandrasekaran, 2024). The effectiveness of this framework hinges on the choice of the scalar hyperparameter $\delta$. A static $\delta$ creates a fundamental trade-off between text quality and watermark detectability. A larger $\delta$ strengthens the watermark and improves detection power but can noticeably degrade generation quality. Conversely, a smaller $\delta$ preserves better fluency yet yields a weak watermark that is difficult to detect reliably. This dilemma is amplified by heterogeneity in next-token logit distributions across models, domains, and languages. Moreover, most existing adaptations operate at the token level, making the bias decision sensitive to local entropy and sampling noise, thereby inducing stepwise fluctuations and unstable detectability (Takezawa et al., 2025; Wouters, 2024; Lee et al., 2024; Wang et al., 2025). A static $\delta$ that appears "optimal" in one setting can be suboptimal or even ineffective in another. These observations motivate a sequence-aware approach that is robust across models and domains.

## 3 METHODOLOGY

We present DynamicBias, a sequence-aware controller that calibrates the bias to the scale of the logits during generation, reducing sensitivity to local noise while retaining a strong signal. It computes a logit-based certainty signal, maintains a sequence-level average, and sets the stepwise bias via a single scaling parameter $\alpha$.

Figure 1 provides an overview of DynamicBias. It replaces the static $\delta$ with a stepwise bias $\delta_t$ calibrated by a sequence-level average of the margin. The watermarked logits are

$$\tilde{\ell}_i^t = \ell_i^t + \delta_t \mathbf{1}\{v_i \in \mathcal{V}_g\}, \qquad i = 1, \ldots, |\mathcal{V}|. \tag{3}$$

Compared to Eq. (1), Eq. (3) replaces the static bias $\delta$ with a stepwise bias $\delta_t$. No other changes are made to hashing, partitioning, or decoding.

## 3.1 CERTAINTY MARGIN

We quantify stepwise certainty using a margin between the top logit and a small pool of next-best logits. Entropy requires a full softmax and can saturate under probability compression, whereas a logit margin avoids softmax and preserves dynamic range. Since watermarking operates in logit space, a logit-based control keeps everything in the same space, reducing scale and/or threshold fragilities and improving stability across models and domains.

Let $\ell^t_{(1)} \geq \ell^t_{(2)} \geq \cdots$ denote the logits sorted in descending order. Define the instantaneous margin

$$m_t = \ell^t_{(1)} - \frac{1}{c-1} \sum_{k=2}^{c} \ell^t_{(k)}. \tag{4}$$

where $c$ is a small integer specifying how many of the next-best logits are averaged. This margin is a stable proxy for certainty: it is less sensitive to tail noise than entropy and requires no access to model internals beyond the logits.

## 3.2 SEQUENCE-AWARE CALIBRATION

To avoid reacting to token-level noise, we maintain a sequence-level average of the margin over the generated steps. Let $t \in \{n+1, \ldots, m\}$ and define the number of tokens generated so far as $K_t = t - n$. Define

$$\bar{m}_t = \frac{1}{K_t} \sum_{s=n+1}^{t} m_s, \tag{5}$$

where $\bar{m}_{n+1} = m_{n+1}$. Equivalently, we can update $\bar{m}_t$ recursively, without storing past margins:

$$\bar{m}_t = \bar{m}_{t-1} + \frac{m_t - \bar{m}_{t-1}}{K_t}. \tag{6}$$

We then set the stepwise bias by a single parameter $\alpha > 0$ as $\delta_t = \alpha \bar{m}_t$. When the current top token is red, a sufficient condition for a flip to green at step $t$ is $\alpha s_t \geq L_r - L_g$, where $s_t$ is the control signal and $L_r$ and $L_g$ are the logits of the top red and top green candidates, respectively. With $s_t = m_t$, this condition often fails in uncertain regions because $m_t$ shrinks precisely when a flip is needed. Using $s_t = \bar{m}_t$ carries confidence from earlier steps, so $\alpha \bar{m}_t \geq L_r - L_g$ holds more frequently and selections are more stable across models.

## 3.3 DIRECTION OF ADAPTATION AND DETECTION

When the model is confident (large $\bar{m}_t$), a given green-list bias induces a smaller distortion of relative probabilities. Our calibration therefore increases $\delta_t$ with $\bar{m}_t$, improving detectability while preserving quality. When the model is uncertain (small $\bar{m}_t$), the bias is correspondingly small, thereby avoiding over-biasing and maintaining fluency.

For detection, we use the standard $z$-score test used in KGW, SWEET, and MorphMark (Kirchenbauer et al., 2024; Lee et al., 2024; Wang et al., 2025).

## 3.4 THEORETICAL ANALYSIS

Under a fixed KGW partition $(\mathcal{V}_g, \mathcal{V}_r)$ at step $t$, let $p_t(i) \propto e^{\ell^t_i}$ and $p^g_t = \sum_{i \in \mathcal{V}_g} p_t(i)$. For a small green-list bias $\delta_t > 0$, the watermarked logits are $\tilde{\ell}^t_i = \ell^t_i + \delta_t \mathbf{1}\{i \in \mathcal{V}_g\}$.

**Lemma 1 (First-order green-mass gain).** *For small $\delta_t$, $\Delta p^g_t \approx \beta_t \delta_t$, $\beta_t = p^g_t(0)\big(1 - p^g_t(0)\big)$, where $p^g_t(0)$ is the green mass at zero bias. (See Appendix A.1 for the proof).*

**Lemma 2 (Quadratic local cost).** *For smooth quality metrics (e.g., KL, Bhattacharyya), the per-step quality loss admits a local quadratic form $QualityLoss_t \approx \kappa_t \delta_t^2, \kappa_t > 0$, and under a local KL proxy one may take $\kappa_t \simeq \frac{1}{2}\beta_t$. (See Appendix A.2 for the proof).*

Let $T$ be the number of generated tokens and $\gamma = |\mathcal{V}_g|/|\mathcal{V}|$ the null green fraction. Since

$$\mathbb{E}[z] = \frac{1}{\sqrt{T\,\gamma(1-\gamma)}} \sum_t (p^g_t - \gamma), \implies \Delta\mathbb{E}[z] \approx \frac{1}{\sqrt{T\,\gamma(1-\gamma)}} \sum_t \Delta p^g_t \propto G(\boldsymbol{\delta}),$$

maximizing the linear surrogate gain $G(\boldsymbol{\delta}) = \sum_t \beta_t \, \delta_t$ under a quadratic cost budget $C(\boldsymbol{\delta}) = \sum_t \kappa_t \, \delta_t^2 \leq B$ approximately increases the expected $z$-score in the small-bias regime. Here, $B > 0$ denotes a global cost budget that upper-bounds the cumulative quadratic loss $C(\boldsymbol{\delta})$.

**Theorem 1 (DynamicBias calibration).** *Introduce a positive weight $\omega$ to trade off gain and cost. DynamicBias sets $\delta_t = \alpha \, \bar{m}_t$, where $\bar{m}_t$ is the sequence-level average of a logit-margin signal and $\alpha > 0$ is a scalar. Define $S = \sum_t \beta_t \, \bar{m}_t, K = \sum_t \kappa_t \bar{m}_t^2$. Then, $G(\alpha) = \alpha S$ and $C(\alpha) = \alpha^2 K$, and the Lagrangian-style objective:*

$$F_\omega(\alpha) = \omega \, S \, \alpha - K \, \alpha^2$$

*is strictly concave in $\alpha$ with the unique maximizer*

$$\alpha^* = \frac{\omega S}{2K} \in (0, \infty), \qquad \frac{\partial \alpha^*}{\partial S} = \frac{\omega}{2K} > 0, \qquad \frac{\partial \alpha^*}{\partial K} = -\frac{\omega S}{2K^2} < 0.$$

*(See Appendix A.3 for the proof).*

As the sequence-level margin contribution $S$ grows (i.e., the model is more confident on average), the optimal scale $\alpha^*$ increases, yielding larger expected detectability for a given cost; if the cost curvature $K$ grows, $\alpha^*$ should decrease.

**Relation to MorphMark.** MorphMark's Theorem 1 establishes token-level monotonicity of the optimal strength with respect to the green-mass signal in a probability-reweighting scheme (Wang et al., 2025). Our analysis instead targets KGW-style additive logit bias with a sequence-level margin controller. Hence, their token-level theorem does not transfer directly to our parameterization (Takezawa et al., 2025; Wouters, 2024). Empirically, we observe trends consistent with the token-level theory while gaining stability from sequence-level averaging.

## 4 EXPERIMENTS

### 4.1 EXPERIMENTAL SETUP

**Datasets.** We evaluate watermarking methods on both open-ended generation and reasoning tasks across multiple languages. (1) C4 and mC4: We use the C4 dataset as prompts for open-ended continuation. For the Japanese and Korean datasets, we utilize the mC4 dataset, which consists of web-scraped content from Common Crawl (Xue et al., 2021). For both C4 and mC4, we randomly sample 500 instances from each test set by following previous work (Kirchenbauer et al., 2024; Guo et al., 2024; Kuditipudi et al., 2024). (2) GSM8K and MGSM: For arithmetic reasoning, we use the GSM8K benchmark of grade-school math word problems. For the Japanese dataset, we employ the MGSM benchmark (Shi et al., 2022), which contains 250 test instances per language. In the MGSM experiments, we use eight training instances as few-shot examples. For the Korean dataset, due to the absence of an original MGSM dataset, we utilize a translated version of GSM8K.[1]

**Evaluation Metrics.** For text quality, we follow prior work (Ren et al., 2024; Kirchenbauer et al., 2024; Guo et al., 2024; Chen et al., 2024; Lu et al., 2024) and report perplexity (PPL) on the C4 and mC4 datasets. For assessing mathematical reasoning capability, we measure accuracy based on the correctness of the model's generated solutions. To evaluate detection performance, we utilize the area under the receiver operating characteristic curve (AUC), a standard metric for binary classification. Due to limited space, we report TPR at FPR$\leq$5% (TPR@5) for detectability in Appendix Tables.

**Baselines. KGW** is a vanilla watermarking method that partitions the LLM vocabulary into "green" and "red" token sets to enable detection of watermarked text (Kirchenbauer et al., 2024). **Unigram** fixes the green set via a fixed hash key within KGW to improve robustness (Zhao et al., 2024). **UPV** (Liu et al., 2024a) proposes a cryptographically secure scheme that enables public verification through an auxiliary detector. We integrate **DynamicBias** into KGW, Unigram, and UPV. We also compare against **SWEET**, which applies watermarking only in high-entropy contexts (Lee et al., 2024), and **MorphMark**, which adaptively scales the bias according to the token-level green probability mass (Wang et al., 2025).

---

[1] `https://huggingface.co/datasets/ChuGyouk/GSM8K-Ko`

Table 1: Performance averaged over four LLMs for each method. +D indicates DynamicBias is applied. **Bold** denotes improvements when DynamicBias is applied. Un-Water indicates no watermarking.

| Method | C4 (PPL ↓ / AUC ↑) | | | | | | | | GSM8K (ACC ↑ / AUC ↑) | | | | | | | |
| | EN | | KR | | JA | | AVG | | EN | | KR | | JA | | AVG | |
| | PPL | AUC | PPL | AUC | PPL | AUC | PPL | AUC | ACC | AUC | ACC | AUC | ACC | AUC | ACC | AUC |
| Un-Water | 4.25 | - | 3.81 | - | 4.14 | - | 4.07 | - | 56.28 | - | 9.39 | - | 8.89 | - | 24.85 | - |
| KGW | 6.52 | 0.991 | 6.34 | 0.960 | 6.75 | 0.971 | 6.79 | 0.978 | 53.47 | 0.848 | 8.19 | 0.861 | 8.28 | 0.884 | 23.31 | 0.864 |
| Unigram | 6.45 | 0.990 | 5.28 | 0.967 | 6.18 | 0.971 | 6.14 | 0.979 | 52.47 | 0.814 | 7.92 | 0.857 | 8.02 | 0.889 | 22.80 | 0.853 |
| UPV | 6.11 | 0.992 | 5.27 | 0.937 | 5.66 | 0.942 | 5.76 | 0.958 | 52.12 | 0.784 | 7.85 | 0.808 | 7.71 | 0.834 | 22.56 | 0.809 |
| SWEET | 6.02 | 0.989 | 5.79 | 0.958 | 6.18 | 0.966 | 6.32 | 0.975 | 53.56 | 0.772 | 8.47 | 0.846 | 8.36 | 0.863 | 23.46 | 0.827 |
| MorphMark | 4.40 | 0.934 | 4.05 | 0.880 | 4.30 | 0.871 | 4.25 | 0.895 | 57.51 | 0.663 | 8.89 | 0.691 | 8.85 | 0.726 | 25.08 | 0.693 |
| KGW+D | **5.32** | **0.996** | **5.43** | **0.983** | **6.02** | **0.985** | **5.80** | **0.989** | 51.44 | **0.966** | 8.12 | **0.949** | 8.47 | **0.958** | 22.68 | **0.958** |
| Unigram+D | **5.23** | **0.993** | **4.97** | **0.980** | **5.75** | **0.978** | **5.50** | **0.986** | 51.21 | **0.946** | 8.30 | **0.909** | 7.77 | **0.941** | 22.43 | **0.932** |
| UPV+D | **5.23** | **0.995** | **5.04** | **0.961** | **5.48** | **0.960** | **5.35** | **0.967** | 47.86 | **0.934** | 7.58 | **0.895** | 7.43 | **0.915** | 20.96 | **0.915** |

**Models and Implementation Details.** We evaluate four LLMs from two families: Llama3.2-3B-Instruct (Grattafiori et al., 2024), Qwen2.5-3B-Instruct (Qwen et al., 2025), Qwen3-1.7B, and Qwen3-4B (Yang et al., 2025). To evaluate PPL, we employ the Qwen-2.5-14B instruction-tuned model. Following prior work, we employ greedy decoding (Chen et al., 2024; Guan et al., 2024; Kirchenbauer et al., 2024). For baseline watermarkings, we apply $\gamma = 0.3$ and $\delta = 3.0$ (Chen et al., 2024). We set the minimum generation length to 50 and the maximum to 200. We apply DynamicBias with a stepwise bias $\delta_t = \alpha \bar{m}_t$ with $\alpha = 0.45$. For the certainty margin, we use $c = 5$. We reset $\bar{m}_t$ for every new prompt to avoid cross-sequence contamination. We use MarkLLM, an open-source toolkit for LLM watermarking, and carefully implement all baselines (Pan et al., 2024). Detailed settings are in Appendix A.4.

### 4.2 PRELIMINARY INVESTIGATION

We first conducted a preliminary study to quantify the relationship between the next-token logit distribution, on which a static bias $\delta$ is applied, and detection performance. Figure 2 shows that settings with small logit gaps achieve very high detectability, whereas settings with large gaps exhibit marked drops in AUC. Using the Unigram watermark, we observed a strong negative correlation between the sequence-averaged logit gap and AUC (Pearson $r = -0.960$). These results indicate that a static bias is not universally optimal: a value tuned for one model or domain can be too weak or too strong in others, yielding inconsistent performance.

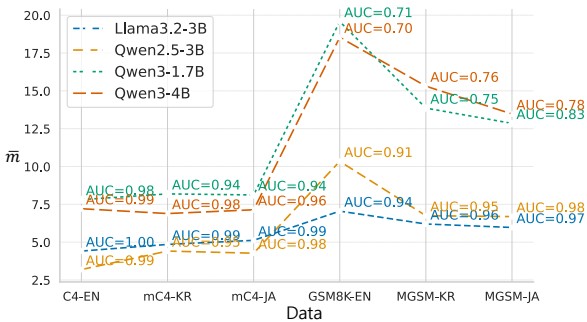

Figure 2: Sequence-averaged logit gap and detectability. The x-axis denotes datasets; the y-axis denotes top logit minus the mean of ranks 2-5.

They support our hypothesis from the introduction that the static bias ignores heterogeneity in logit distributions across models, domains, and languages, and therefore does not generalize reliably. With a static bias $\delta$, small-gap settings often yield higher detectability because the same additive shift moves relatively more mass from the red to the green partition; however, this also risks quality degradation. Conversely, in large-gap settings, the shift causes less relative distortion, tending to preserve quality but weakening the watermark signal and reducing AUC.

### 4.3 OVERALL PERFORMANCE

Table 1 shows results averaged over four LLMs. Per-model results are in Appendix B.1. Incorporating DynamicBias into KGW, Unigram, and UPV consistently improves detectability across all baselines. For C4 and mC4, AUC increases with DynamicBias while perplexity decreases. For GSM8K and MGSM, applying DynamicBias to KGW, Unigram, and UPV yields larger AUC gains,

with a slight drop in accuracy. SWEET achieves higher AUC than MorphMark on C4 and mC4 but struggles on GSM8K and MGSM, whereas MorphMark shows weaker detectability overall. While DynamicBias improves detection performance without sacrificing text quality, its gains are smaller in non-English languages. We think this is because of language-specific differences in logit margins. Additional experimental results comparing our method with the method that uses an auxiliary model for learning bias are provided in Appendix B.2.

## 4.4 PERFORMANCE ON ROBUSTNESS

Because paraphrasing can diminish watermark signals, we evaluate three paraphrasing attacks to assess robustness. First, WordDel randomly deletes 50% of words from the text.

Second, WordSub randomly replaces 50% of words with synonyms from WordNet. Third, Dipper rewrites the text with a specialized paraphrasing model. Table 2 shows the results using Qwen3-4B. As shown, DynamicBias improves detectability for KGW, Unigram, and UPV under all attacks. These results indicate that DynamicBias is more robust and therefore more practical and reliable.

Table 2: AUC under attacks on C4 and GSM8K.

| Method | C4 (AUC ↑) | | | | GSM8K (AUC ↑) | | | |
|---|---|---|---|---|---|---|---|---|
| | Original | WordDel | WordSub | Dipper | Original | WordDel | WordSub | Dipper |
| KGW | 0.997 | 0.934 | 0.825 | 0.787 | 0.824 | 0.669 | 0.581 | 0.739 |
| Unigram | 0.995 | 0.983 | 0.921 | 0.914 | 0.698 | 0.607 | 0.583 | 0.595 |
| UPV | 0.991 | 0.932 | 0.852 | 0.804 | 0.668 | 0.575 | 0.551 | 0.417 |
| KGW+D | **0.998** | **0.939** | **0.841** | **0.794** | **0.974** | **0.793** | **0.681** | **0.752** |
| Unigram+D | **0.997** | **0.988** | **0.942** | **0.918** | **0.916** | **0.804** | **0.766** | **0.655** |
| UPV+D | **0.995** | **0.959** | **0.855** | **0.808** | **0.900** | **0.689** | **0.639** | **0.445** |

## 4.5 PERFORMANCE ON TEXT QUALITY AND IN LOW-ENTROPY SCENARIOS

To evaluate text quality beyond PPL and in low-entropy settings, we conduct experiments on CNN/DM for document summarization (Hermann et al., 2015) and HumanEval for code generation (Chen et al., 2021). For summarization, we use ROUGE-L (R-L) (Lin, 2004) and BERTScore (BS) (Zhang et al., 2020). For code generation, we use Pass@1. We keep the same parameters as in the main study. Table 3 shows the results using Qwen3-4B. DynamicBias substantially improves detectability with acceptable drops in R-L, BS, and Pass@1. These small decreases reflect the quality and detectability trade-off, while AUC increases markedly. Details on the datasets and settings are in Appendix C.

Table 3: CNN/DM summarization and HumanEval code generation.

| Method | CNN/DM | | | HumanEval | |
|---|---|---|---|---|---|
| | R-L↑ | BS↑ | AUC↑ | Pass@1↑ | AUC↑ |
| KGW | 28.33 | 63.57 | 0.908 | 0.3598 | 0.602 |
| Unigram | 29.56 | 64.87 | 0.896 | 0.3841 | 0.558 |
| UPV | 29.47 | 65.02 | 0.901 | 0.3354 | 0.601 |
| SWEET | 29.21 | 64.41 | 0.752 | 0.3841 | 0.504 |
| MorphMark | 29.69 | 64.72 | 0.714 | 0.3902 | 0.525 |
| KGW+D | 27.00 | 63.06 | **0.996** | 0.3049 | **0.818** |
| Unigram+D | 29.43 | 64.63 | **0.985** | 0.3537 | **0.675** |
| UPV+D | 28.35 | 64.44 | **0.985** | 0.3232 | **0.788** |

## 4.6 BIN-WISE ANALYSIS FOR DETECTION PERFORMANCE

Table 4 shows the results for the bin-wise analysis on the C4 and mC4 datasets. By using Qwen3-4B, we group generations by sequence length and compute detection performance per bin, confirming that DynamicBias maintains strong detection even for short outputs.

Table 4: AUC and TPR@5 across token-range segments.

| Data | Method | 0–25 | | 25–50 | | 50–75 | | 75–100 | |
|---|---|---|---|---|---|---|---|---|---|
| | | AUC | TPR@5 | AUC | TPR@5 | AUC | TPR@5 | AUC | TPR@5 |
| C4-EN | KGW | 0.93 | 0.71 | 0.98 | 0.89 | 0.99 | 0.94 | 0.99 | 0.98 |
| | KGW+D | **0.95** | **0.79** | **0.98** | **0.92** | **0.99** | **0.97** | **1.0** | **0.99** |
| mC4-KR | KGW | 0.89 | 0.59 | 0.94 | 0.75 | 0.96 | 0.89 | 0.97 | 0.91 |
| | KGW+D | **0.93** | **0.68** | **0.96** | **0.82** | **0.98** | **0.91** | **0.99** | **0.95** |
| mC4-JA | KGW | 0.85 | 0.50 | 0.92 | 0.78 | 0.95 | 0.84 | 0.96 | 0.89 |
| | KGW+D | **0.89** | **0.59** | **0.95** | **0.84** | **0.97** | **0.90** | **1.0** | **1.0** |

## 4.7 ABLATION STUDY FOR $\alpha$ AND $\gamma$

In this section, we ablate the hyper-parameters of DynamicBias, including $\alpha$ for scaling the bias and $\gamma$ for the green vocabulary proportion. A larger $\alpha$ applies a stronger bias, whereas a smaller $\alpha$ applies a weaker bias. A larger $\gamma$ indicates a higher proportion of green tokens, whereas a smaller $\gamma$ denotes a lower green proportion. Figure 3 (a) shows the results of varying $\alpha$ while keeping other settings fixed. As $\alpha$ increases, the $z$-score on C4 grows almost linearly and PPL increases slowly. On GSM8K, the $z$-score also rises with $\alpha$, while accuracy begins to decrease once $\alpha$ becomes large. Quantitatively, the $z$-score scales nearly linearly with $\alpha$ (C4: $R^2$=0.9992, GSM8K: $R^2$=0.9747),

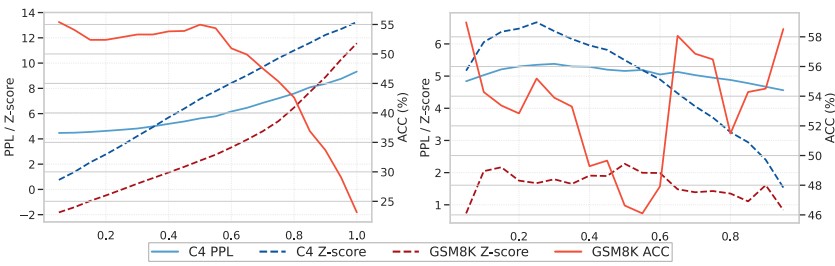

Figure 3: Parameter ablation study of DynamicBias using Qwen3-4B. DynamicBias is applied to KGW. In (a), we ablate $\alpha$. In (b), we ablate $\gamma$. Scores are averaged within each dataset.

whereas quality exhibits mild quadratic curvature (PPL: $R^2=0.9991$, ACC: $R^2=0.9574$), supporting the small-bias linear-gain and low-curvature cost assumptions used in our calibration analysis.

Figure 3 (b) shows the results of varying $\gamma$. On C4, the z-score is higher at small $\gamma$ and then drops as $\gamma$ grows, while PPL remains relatively stable. On GSM8K, which contains many numbers and requires math reasoning, very low $\gamma$ (e.g., 0.1) makes the watermark less responsive and yields a very low z-score. As $\gamma$ increases, more tokens are steered into the green set, detectability improves, and the z-score rises, while accuracy drops slightly due to increased watermark responsiveness. Once $\gamma$ exceeds about 0.6, many numeric tokens fall into the green set, reducing distortion on key tokens; accuracy rises again, but AUC decreases as the signal becomes diluted.

Therefore, $\alpha$ smoothly controls watermark strength and should be set to a moderate value, whereas $\gamma$ should remain small to avoid diluting the signal.

## 4.8 Ablation Study for $c$ and Sequence-Aware Calibration

Table 5: Effect of the pool size $c$ in the margin definition.

| $c$ | C4 | | GSM8K | | HumanEval | |
|---|---|---|---|---|---|---|
| | PPL↓ | AUC↑ | ACC↑ | AUC↑ | Pass@1↑ | AUC↑ |
| 2 | **5.10** | 0.983 | 23.70 | 0.883 | **0.317** | 0.738 |
| 5 | 5.79 | 0.994 | **25.09** | 0.940 | 0.305 | 0.818 |
| 10 | 6.52 | **0.996** | 24.79 | **0.975** | 0.287 | **0.853** |

We vary the pool size $c$ in the margin definition by using Top-2 ($c = 2$), Top-2-5 ($c = 5$), and Top-2-10 ($c = 10$). Table 5 shows the results using Qwen3-4B, averaged over English, Japanese, and Korean. Per-language results are in Appendix B.3. On C4 and HumanEval, AUC increases as $c$ grows, while PPL and Pass@1 degrade moderately. On GSM8K, AUC also increases with $c$, and accuracy is largely stable, with a mild peak around $c = 5$ on average. This trend is expected: as $c$ grows, the next-best pool includes lower-ranked logits, the averaged competitor score declines, the margin becomes larger, and the calibrated bias increases accordingly. Thus, averaging over a larger set of next-best logits reduces variance in the certainty estimate and yields a steadier bias schedule, improving detectability. However, very large $c$ values slightly inflate PPL and can nudge accuracy down in some languages.

Table 6: Performance of sequence-aware calibration and token-level DynamicBias.

| Scope | C4 | | GSM8K | |
|---|---|---|---|---|
| | PPL↓ | AUC↑ | ACC↑ | AUC↑ |
| Morphmark ($k_{linear} = 1.55$) | 4.57 | 0.886 | 26.21 | 0.605 |
| Morphmark ($k_{linear} = 9.0$) | 5.42 | 0.985 | 26.05 | 0.747 |
| Token | **4.75** | 0.894 | **26.61** | 0.716 |
| Sequence | 5.79 | **0.994** | 25.09 | **0.940** |

Table 6 shows the effect of sequence-aware calibration versus a token-level variant using Qwen3-4B, averaged over English, Japanese, and Korean. The token-level variant skips sequence-aware calibration (Sec. 3.2) and uses $\delta_t = \alpha m_t$. We also include results obtained by increasing the MorphMark detection parameter, which is $k_{linear}$, calibrated on the C4 dataset. MorphMark shows limited gains in detection performance and struggles on GSM8K. Sequence-aware calibration delivers higher AUC at a small cost in PPL, with minimal accuracy change. These results are consistent with the intuition that sequence-level averaging increases the effective bias when it matters, making $\alpha \bar{m}_t \geq L_r - L_g$ more often satisfied and yielding green selections. Per-language results and detailed tuning results are in Appendix B.4.

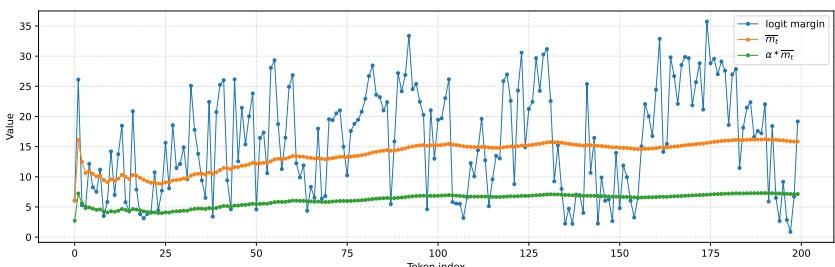

Figure 4: Case study on GSM8K. (a) shows KGW and (b) shows KGW+D. The black text represents the input prompt. green and red highlights mark tokens in the green and red lists, respectively.

Figure 5: Visualization of the the Top-2-5 logit margin ($c = 5$), $\bar{m}_t$, and its scaled version $\alpha\bar{m}_t$ for DynamicBias in (b) of Figure 4. The x-axis indicates the token index and the y-axis shows their values.

### 4.9 RUNTIME COMPARISON

In addition to text quality and detection performance, we compare the runtime overhead of different watermarking methods. We measure both generation and detection time using NVIDIA A6000 GPU. We sampled 100 instances from the C4 dataset by generating 200 tokens for each instance. Table 7 shows the results. DynamicBias does not significantly increase generation time compared to the baseline methods. At detection time, it incurs almost no additional overhead because it does not require computing token entropies or querying the model.

Table 7: Latency comparison across watermarking methods and models.

| Model | Method | Gen. Time (token/ms) | Gen. Time (data/s) | Detect. Time (data/ms) |
|---|---|---|---|---|
| | KGW | **26.28** | **5.25** | **24.9** |
| | KGW + D | 26.41 | 5.28 | 25.0 |
| Llama 3.2 | UPV | 28.61 | 5.71 | 38.5 |
| -3B | UPV + D | 28.74 | 5.74 | 38.9 |
| | SWEET | 26.37 | 5.27 | 133.6 |
| | Morphmark | 26.64 | 5.28 | 26.8 |
| | KGW | **34.93** | **6.97** | **26.2** |
| | KGW + D | 35.17 | 7.03 | 26.3 |
| Qwen 3 | UPV | 37.92 | 7.57 | 44.1 |
| -4B | UPV + D | 38.03 | 7.61 | 44.6 |
| | SWEET | 35.18 | 7.01 | 163.4 |
| | Morphmark | 35.54 | 7.06 | 28.1 |

### 4.10 CASE STUDY

We present a case study using Qwen3-4B between KGW and KGW+DynamicBias (KGW+D) on a GSM8K example, using identical parameter settings. Figure 4 shows the results. We observe that KGW+D generates more green tokens than KGW, resulting in higher detectability and greater robustness to paraphrasing attacks. Appendix B.5 includes the average number of green tokens across datasets. Figure 5 shows the evolution of $\delta_t$ over token steps. Although the second token spikes, the averaged and scaled margins remain stable.

### 4.11 DECODING STRATEGY

We test whether DynamicBias remains effective under different sampling parameters at the decoding stage. We consider top-$p$ sampling at 0.95 with temperature 0.7. Table 8 shows the results averaged over four LLMs. DynamicBias achieves a superior balance between text quality and detectability. Per-model results are in Appendix B.6.

Table 8: Performance averaged over four LLMs with top-$p$ sampling.

| Method | C4 (PPL ↓ / AUC ↑) | | | | | | | | GSM8K (ACC ↑ / AUC ↑) | | | | | | | |
|---|---|---|---|---|---|---|---|---|---|---|---|---|---|---|---|---|
| | EN | | KR | | JA | | AVG | | EN | | KR | | JA | | AVG | |
| | PPL | AUC | PPL | AUC | PPL | AUC | PPL | AUC | ACC | AUC | ACC | AUC | ACC | AUC | ACC | AUC |
| Un-Water | 4.58 | - | 4.31 | - | 4.79 | - | 4.56 | - | 56.43 | - | 9.34 | - | 8.78 | - | 24.85 | - |
| KGW | 7.23 | 0.992 | 7.89 | 0.964 | 8.23 | 0.975 | 7.78 | 0.977 | 52.60 | 0.849 | 8.38 | 0.863 | 8.13 | 0.888 | 23.04 | 0.867 |
| Unigram | 7.17 | 0.990 | 6.40 | 0.972 | 7.62 | 0.973 | 7.06 | 0.978 | 52.12 | 0.813 | 7.62 | 0.858 | 7.62 | 0.897 | 22.45 | 0.856 |
| UPV | 6.67 | 0.992 | 6.02 | 0.941 | 6.57 | 0.949 | 6.42 | 0.961 | 51.48 | 0.785 | 7.83 | 0.813 | 7.66 | 0.841 | 22.32 | 0.813 |
| SWEET | 6.59 | 0.990 | 7.27 | 0.959 | 7.55 | 0.968 | 7.13 | 0.972 | 53.47 | 0.769 | 8.61 | 0.842 | 8.17 | 0.863 | 23.42 | 0.825 |
| MorphMark | 4.70 | 0.933 | 4.59 | 0.882 | 4.89 | 0.877 | 4.73 | 0.897 | 56.94 | 0.665 | 8.68 | 0.686 | 8.83 | 0.718 | 24.82 | 0.690 |
| KGW+D | **5.68** | **0.997** | **6.48** | **0.985** | **7.03** | **0.987** | **6.40** | **0.990** | 51.33 | **0.966** | 8.25 | **0.952** | **8.23** | **0.962** | 22.60 | **0.960** |
| Unigram+D | **5.65** | 0.992 | **6.17** | 0.983 | **7.03** | 0.979 | **6.28** | 0.985 | 50.59 | **0.949** | 8.24 | **0.911** | 7.70 | **0.946** | 22.17 | **0.935** |
| UPV+D | **5.67** | 0.995 | **5.48** | 0.959 | **6.29** | 0.959 | **5.93** | 0.971 | 47.99 | **0.912** | 7.49 | **0.873** | 7.68 | **0.900** | 21.05 | **0.895** |

## 5 RELATED WORK

Early work framed AI-generated text detection as binary classification over finalized text (Jawahar et al., 2020; Mitchell et al., 2023), but performance degrades as LLM fluency and paraphrasing improve. Decoding-based watermarking instead perturbs logits during generation to embed a statistical signature without retraining.

KGW partitions the vocabulary into "green" and "red" sets via a prefix hash and adds a static bias to "green" logits (Kirchenbauer et al., 2024). Variants strengthen robustness by fixing the green set with a hash key (Zhao et al., 2024), imposing mutual-exclusivity constraints (Chen et al., 2024), and enabling public verification with an auxiliary detector (Liu et al., 2024a). Despite progress, static bias schemes face a trade-off between text quality and detectability under model, domain, and language shifts (Guo et al., 2024; Ren et al., 2024; Wu & Chandrasekaran, 2024). Unbiased watermarking preserves the expected sampling distribution to protect quality (Kuditipudi et al., 2024; Hu et al., 2024; Wu et al., 2024), but current implementations often reduce robustness or add nontrivial overhead. While gating a static bias by entropy to target high-entropy contexts (Lee et al., 2024) and scaling bias strength using the token-level cumulative probability mass on the "green" list (Wang et al., 2025) have been studied, these approaches do not explicitly consider sequence-level variability, limiting stability across settings. Paraphrasing and light edits can weaken watermark signals. Previous works show AUC drops for static bias methods under deletion, substitution, and model-based rewrites (Guo et al., 2024; Ren et al., 2024; Wu & Chandrasekaran, 2024). Approaches that rely on auxiliary detectors or model access can improve robustness but reduce the simplicity of model-independent deployment (Liu et al., 2024a).

DynamicBias is a sequence-aware calibration approach for vocabulary-partition watermarks: it replaces the static bias with one calibrated by a sequence-level margin, leaving hashing, partitioning, and detection unchanged. Unlike unbiased methods, it retains the standard detector; unlike entropy-gated schemes, it does not skip low-entropy tokens; unlike token-level scaling, it aggregates certainty over the sequence to reduce stepwise variance. Our theoretical and empirical results show that this improves detectability while maintaining competitive quality and increases paraphrase robustness.

## 6 CONCLUSION

We introduced DynamicBias, a sequence-aware controller that replaces the static bias with one calibrated to the scale of next-token logits. The method computes a margin between the top logit and a small pool of next-best logits, averages it over the sequence, and sets a stepwise bias via a single scaling parameter $\alpha$. Across four LLMs and three languages, DynamicBias improves detectability while maintaining text quality, and it remains effective under paraphrasing attacks. Furthermore, DynamicBias works across different models and domains and integrates easily with vocabulary-partition watermarks. In the future, we will extend DynamicBias beyond vocabulary partitioning and explore learning $\alpha$.

## 7 ETHICS STATEMENT

All datasets used in this study are publicly available resources from the research community. We did not create or collect any new sensitive content. Beyond introducing a new watermarking framework, our work also provides theoretical analysis to help the community better understand vocabulary-partition watermarking. We believe this contributes positively to the development of safer and more robust LLMs.

## 8 REPRODUCIBILITY STATEMENT

All four models used in our experiments (`Llama3.2-3B-Instruct`, `Qwen2.5-3B-Instruct`, `Qwen3-1.7B`, and `Qwen3-4B`) are open-source and publicly available. To facilitate replication, we will release our source code and results, with detailed explanations, to fully reproduce our results. We note that our methods are applicable to models that provide access to top-$k$ log probabilities.

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

# A PROOF

## A.1 THE PROOF FOR LEMMA 1

Fix a step $t$. Define $G = \sum_{i \in \mathcal{V}_g} e^{\ell_i^t}$ and $R = \sum_{i \notin \mathcal{V}_g} e^{\ell_i^t}$. Adding a bias $\delta_t$ to green logits yields $\tilde{\ell}_i^t = \ell_i^t + \delta_t \mathbf{1}\{i \in \mathcal{V}_g\}$. The next-token probability under softmax is as follows:

$$p_t(i; \delta_t) = \frac{\exp(\tilde{\ell}_i^t)}{\sum_j \exp(\tilde{\ell}_j^t)} = \frac{\exp\left(\ell_i^t + \delta_t \mathbf{1}\{i \in \mathcal{V}_g\}\right)}{\sum_{j \in \mathcal{V}_g} \exp(\ell_j^t + \delta_t) + \sum_{j \in \mathcal{V}_r} \exp(\ell_j^t)} = \frac{\exp\left(\ell_i^t + \delta_t \mathbf{1}\{i \in \mathcal{V}_g\}\right)}{e^{\delta_t} G + R},$$
(7)

After adding $\delta_t$ to green logits,

$$p_t^g(\delta_t) = \frac{e^{\delta_t} G}{e^{\delta_t} G + R}.$$
(8)

Differentiate:

$$\frac{\partial p_t^g(\delta_t)}{\partial \delta_t} = \frac{e^{\delta_t} G R}{(e^{\delta_t} G + R)^2}.$$
(9)

At $\delta_t = 0$, the derivative at 0 equals $\frac{GR}{(G+R)^2} = \frac{G}{G+R}\left(1 - \frac{G}{G+R}\right) = p_t^g(0)\left(1 - p_t^g(0)\right) \equiv \beta_t$.

A first-order Taylor expansion gives $p_t^g(\delta_t) \approx p_t^g(0) + \beta_t \delta_t$, i.e., $\Delta p_t^g \approx \beta_t \delta_t$.

## A.2 THE PROOF FOR LEMMA 2 (KL CASE)

Fix a step $t$. Define $G = \sum_{i \in \mathcal{V}_g} e^{\ell_i^t}$ and $R = \sum_{i \notin \mathcal{V}_g} e^{\ell_i^t}$, $Z = G + R$, $p_t^g(0) = \frac{G}{Z}$, and $\beta_t = p_t^g(0)\left(1 - p_t^g(0)\right)$.

Let $p_t(i; 0) = \frac{e^{\ell_i^t}}{Z}$ be the pre-bias next-token distribution and let $q_t(i; \delta_t) = \frac{e^{\ell_i^t + \delta_t \mathbf{1}\{i \in \mathcal{V}_g\}}}{e^{\delta_t} G + R}$ be the distribution after adding a green-only bias $\delta_t$.

The per-step KL divergence is as follows:

$$\begin{aligned}
\mathrm{KL}_t &= \sum_i p_t(i; 0) \log \frac{p_t(i; 0)}{q_t(i; \delta_t)} \\
&= \sum_i \frac{e^{\ell_i^t}}{Z}\left[\left(\ell_i^t - \log Z\right) - \left(\ell_i^t + \delta_t \mathbf{1}\{i \in \mathcal{V}_g\} - \log(e^{\delta_t} G + R)\right)\right] \\
&= \sum_i \frac{e^{\ell_i^t}}{Z}\left[-\delta_t \mathbf{1}\{i \in \mathcal{V}_g\} + \log(e^{\delta_t} G + R) - \log Z\right].
\end{aligned}$$
(10)

Using $\sum_i \frac{e^{\ell_i^t}}{Z} \mathbf{1}\{i \in \mathcal{V}_g\} = p_t^g(0) = G/Z$ and $\sum_i \frac{e^{\ell_i^t}}{Z} = 1$, this simplifies to the scalar function as follows:

$$\mathrm{KL}_t = -p_t^g(0)\delta_t + \log\left(\frac{e^{\delta_t} G + R}{Z}\right) = -\frac{G}{Z}\delta_t + \log\left(\frac{e^{\delta_t} G + R}{G + R}\right).$$
(11)

Define $D(\delta_t) := \frac{e^{\delta_t} G + R}{G + R} = \frac{G}{Z} e^{\delta_t} + \frac{R}{Z} = p_t^g(0) e^{\delta_t} + \left(1 - p_t^g(0)\right)$.

A second-order Taylor expansion and using $\log(1 + x) = x - \frac{1}{2}x^2 + O(x^3)$, at $\delta_t = 0$ gives

$$\begin{aligned}
D(\delta_t) &= 1 + p_t^g(0)\delta_t + \tfrac{1}{2} p_t^g(0)\delta_t^2 + O(\delta_t^3) \\
\log D(\delta_t) &= p_t^g(0)\delta_t + \tfrac{1}{2} p_t^g(0)\left(1 - p_t^g(0)\right)\delta_t^2 + O(\delta_t^3).
\end{aligned}$$
(12)

Therefore

$$\mathrm{KL}_t = \underbrace{p_t^g(0)\delta_t - p_t^g(0)\delta_t}_{= 0} + \tfrac{1}{2}p_t^g(0)\left(1 - p_t^g(0)\right)\delta_t^2 + O(\delta_t^3) = \tfrac{1}{2}\beta_t \delta_t^2 + O(\delta_t^3),$$
(13)

which shows that the local quadratic coefficient is $\kappa_t = \tfrac{1}{2}\beta_t > 0$.

### A.3 The proof for Theorem 1

Let $\delta_t = \alpha \bar{m}_t$ with $\bar{m}_t \geq 0$, and define $S = \sum_t \beta_t \bar{m}_t \quad (\beta_t \geq 0), K = \sum_t \kappa_t \bar{m}_t^2 \quad (\kappa_t > 0)$. Then, the gain and cost surrogates become $G(\alpha) = \alpha S$ and $C(\alpha) = \alpha^2 K$, so the objective is as follows:

$$F_\omega(\alpha) = \omega S \alpha - K \alpha^2, \omega > 0. \tag{14}$$

Compute derivatives:

$$F'_\omega(\alpha) = \omega S - 2K\alpha, F''_\omega(\alpha) = -2K. \tag{15}$$

Since $K = \sum_t \kappa_t \bar{m}_t^2 > 0$, we have $F''_\omega(\alpha) = -2K < 0$, so $F_\omega$ is strictly concave in $\alpha$. Setting $F'_\omega(\alpha) = 0$ gives the unique critical point $\alpha^* = \omega S/(2K)$, which is therefore the unique global maximizer. Moreover,

$$\frac{\partial \alpha^*}{\partial S} = \frac{\omega}{2K} > 0, \frac{\partial \alpha^*}{\partial K} = -\frac{\omega S}{2K^2} < 0, \tag{16}$$

showing that the optimal calibration increases with $S$ and decreases with $K$.

**Assumptions and scope.** (i) Small-bias regime so that the first-/second-order expansions are accurate; (ii) linearity of expectation (or weak dependence) in the $z$-statistic's numerator; (iii) a fixed KGW partition; (iv) under a local KL proxy, $\kappa_t \simeq \frac{1}{2} \beta_t$ is common, which can make the reward–cost ratio $r_t = \beta_t/\kappa_t$ approximately constant; empirical comparisons then become especially informative.

### A.4 Implementation Details

For the watermarking methods, we set the SWEET entropy threshold to 0.9. For UPV, we use a 16 bit-number and set $\sigma$ to 0.01. For MorphMark, we use the exponential type with parameters $k_{exp}$=1.30 and $p_0$=0.15. Note that we extensively tuned MorphMark's hyper-parameters.

DynamicBias adds only two lightweight operations per step: (1) compute the certainty margin from the top-$c$ logits and update the sequence-level average $\bar{m}_t$ (margin $O(c)$, update $O(1)$); (2) set the calibrated bias $\delta_t = \alpha \bar{m}_t$. No extra forward passes, gradients, or model-specific training are required.

## B Additional Experiments

### B.1 Overall Performance

Since the main text includes only results averaged over four LLMs, we provide per-model results. Tables 9 and 10 show the results on C4/mC4 and GSM8K/MGSM. We observe that integrating DynamicBias is effective across models.

### B.2 Comparison to using an Auxiliary Model

We also compare DynamicBias with the auxiliary-model approach of TS (Huo et al., 2024), which is not model-independent. For a fair comparison, we employ the OPT-1.3B (Zhang et al., 2022) and Llama2-13B-Instruct (Touvron et al., 2023) models. Table 11 shows the results. Under the TS watermarking configuration with temperature=1 and top-k=50, DynamicBias consistently outperforms TS.

### B.3 Effect of the pool size $c$

Since the main text includes only results averaged over languages, we provide per-language results. Table 12 shows the results on C4/mC4 and GSM8K/MGSM.

### B.4 Sequence-Aware Calibration

Since the main text includes only results averaged over languages, we provide per-language results. Table 13 shows the results on C4/mC4 and GSM8K/MGSM using Qwen3-4B. We observed that

Table 9: Performance of each method on C4 and mC4 across languages with averaged scores (AVG).

| Model | Method | EN | | | KR | | | JA | | | AVG | | |
|---|---|---|---|---|---|---|---|---|---|---|---|---|---|
| | | PPL | AUC | TPR@5 | PPL | AUC | TPR@5 | PPL | AUC | TPR@5 | PPL | AUC | TPR@5 |
| Llama3.2 -3B | Un-Water | 3.49 | - | - | 3.86 | - | - | 4.53 | - | - | 3.96 | - | - |
| | KGW | 5.83 | 0.996 | 0.990 | 7.48 | 0.987 | 0.966 | 7.76 | 0.988 | 0.968 | 7.02 | 0.990 | 0.975 |
| | Unigram | 6.55 | 0.998 | 0.996 | 4.86 | 0.988 | 0.962 | 6.76 | 0.995 | 0.986 | 6.06 | 0.994 | 0.981 |
| | UPV | 5.36 | 1.000 | 0.998 | 5.70 | 0.960 | 0.858 | 6.31 | 0.968 | 0.924 | 5.79 | 0.976 | 0.926 |
| | SWEET | 5.25 | 0.996 | 0.990 | 7.07 | 0.989 | 0.966 | 7.28 | 0.989 | 0.972 | 6.53 | 0.991 | 0.976 |
| | MorphMark | 3.62 | 0.974 | 0.852 | 4.65 | 0.898 | 0.672 | 4.87 | 0.899 | 0.650 | 4.38 | 0.924 | 0.725 |
| | KGW+D | **4.62** | **0.999** | **0.996** | **6.02** | **0.990** | **0.968** | **6.62** | **0.989** | **0.968** | **5.75** | **0.993** | **0.977** |
| | Unigram+D | **4.72** | **1.000** | **0.998** | 5.15 | **0.989** | **0.966** | **6.34** | 0.991 | 0.968 | **5.40** | 0.993 | 0.977 |
| | UPV+D | **4.50** | 0.998 | 0.992 | **5.42** | 0.947 | 0.784 | **6.04** | 0.960 | 0.884 | **5.32** | 0.968 | 0.886 |
| Qwen2.5 -3B | Un-Water | 3.93 | - | - | 3.30 | - | - | 3.26 | - | - | 3.50 | - | - |
| | KGW | 8.96 | 0.992 | 0.982 | 7.22 | 0.968 | 0.930 | 7.24 | 0.989 | 0.976 | 7.81 | 0.983 | 0.962 |
| | Unigram | 8.03 | 0.990 | 0.964 | 6.55 | 0.955 | 0.892 | 7.18 | 0.980 | 0.960 | 7.25 | 0.975 | 0.939 |
| | UPV | 7.82 | 0.991 | 0.978 | 5.56 | 0.943 | 0.874 | 5.62 | 0.975 | 0.950 | 6.33 | 0.970 | 0.934 |
| | SWEET | 8.28 | 0.988 | 0.974 | 6.27 | 0.949 | 0.904 | 6.46 | 0.979 | 0.954 | 7.00 | 0.972 | 0.944 |
| | MorphMark | 4.29 | 0.968 | 0.952 | 3.41 | 0.887 | 0.724 | 3.48 | 0.914 | 0.762 | 3.73 | 0.923 | 0.813 |
| | KGW+D | **5.02** | **0.999** | **0.998** | **4.97** | **0.990** | **0.970** | **5.19** | **0.995** | **0.986** | **5.06** | **0.995** | **0.985** |
| | Unigram+D | **4.86** | **0.995** | **0.980** | **4.81** | **0.984** | **0.960** | **5.17** | **0.984** | **0.966** | **4.95** | **0.987** | **0.969** |
| | UPV+D | **4.95** | **0.997** | **0.990** | **4.65** | **0.974** | **0.904** | **4.72** | **0.985** | **0.954** | **4.77** | **0.985** | **0.949** |
| Qwen3 -1.7B | Un-Water | 5.11 | - | - | 3.80 | - | - | 4.06 | - | - | 4.32 | - | - |
| | KGW | 5.99 | 0.979 | 0.960 | 5.10 | 0.906 | 0.830 | 5.67 | 0.929 | 0.868 | 5.59 | 0.938 | 0.886 |
| | Unigram | 5.96 | 0.976 | 0.942 | 4.40 | 0.940 | 0.856 | 5.09 | 0.945 | 0.798 | 5.15 | 0.954 | 0.865 |
| | UPV | 5.98 | 0.985 | 0.952 | 4.55 | 0.892 | 0.760 | 5.05 | 0.897 | 0.766 | 5.19 | 0.925 | 0.826 |
| | SWEET | 5.65 | 0.977 | 0.940 | 4.73 | 0.910 | 0.824 | 5.19 | 0.919 | 0.844 | 5.19 | 0.935 | 0.869 |
| | MorphMark | 5.19 | 0.899 | 0.584 | 3.79 | 0.839 | 0.454 | 3.98 | 0.804 | 0.724 | 4.32 | 0.847 | 0.587 |
| | KGW+D | 6.25 | **0.988** | **0.970** | 5.15 | **0.956** | **0.906** | 5.89 | **0.966** | **0.928** | 5.76 | **0.970** | **0.935** |
| | Unigram+D | 6.03 | **0.982** | **0.952** | 4.57 | **0.961** | **0.884** | 5.37 | **0.967** | **0.892** | 5.32 | **0.970** | **0.909** |
| | UPV+D | 6.13 | **0.991** | **0.956** | 4.82 | **0.943** | **0.858** | 5.36 | **0.934** | **0.842** | 5.44 | **0.956** | **0.885** |
| Qwen3 -4B | Un-Water | 4.48 | - | - | 4.27 | - | - | 4.7 | - | - | 4.48 | - | - |
| | KGW | 5.29 | 0.997 | 0.994 | 5.55 | 0.978 | 0.950 | 6.31 | 0.977 | 0.962 | 5.72 | 0.984 | 0.969 |
| | Unigram | 5.26 | 0.995 | 0.990 | 5.29 | 0.984 | 0.948 | 5.70 | 0.964 | 0.824 | 5.42 | 0.981 | 0.921 |
| | UPV | 5.27 | 0.991 | 0.984 | 5.26 | 0.953 | 0.880 | 5.65 | 0.929 | 0.806 | 5.39 | 0.958 | 0.890 |
| | SWEET | 4.90 | 0.997 | 0.980 | 5.09 | 0.984 | 0.960 | 5.77 | 0.978 | 0.946 | 5.25 | 0.986 | 0.962 |
| | MorphMark | 4.5 | 0.895 | 0.522 | 4.33 | 0.898 | 0.542 | 4.88 | 0.865 | 0.456 | 4.57 | 0.886 | 0.507 |
| | KGW+D | 5.38 | **0.998** | **0.996** | 5.59 | **0.994** | **0.984** | 6.39 | **0.990** | **0.990** | 5.79 | **0.994** | **0.990** |
| | Unigram+D | 5.31 | **0.997** | **0.996** | 5.34 | **0.986** | **0.952** | 6.10 | **0.971** | **0.850** | 5.58 | **0.985** | **0.933** |
| | UPV+D | 5.34 | **0.995** | **0.996** | 5.26 | **0.979** | **0.942** | 5.78 | **0.963** | **0.866** | 5.46 | **0.979** | **0.935** |

sequence-aware calibration delivers higher detectability across languages than the token-level variant. Figure 6 presents the parameter-tuning results for MorphMark.

### B.5    NUMBER OF GREEN TOKENS GENERATED

Table 14 shows the average number of green tokens on C4 and GSM8K using Qwen3-4B. Because WordSub replaces words using WordNet, it often increases the total token count; accordingly, counts are higher under this attack. We also observe that applying DynamicBias increases the number of green tokens generated.

### B.6    DECODING STRATEGY

We provide per-model results. Tables 15 and 16 show the results on C4/mC4 and GSM8K/MGSM using top-$p$ decoding strategy with temperature.

Table 10: Performance of each method on GSM8K and MGSM across languages with averaged scores (AVG).

| Model | Method | EN | | | KR | | | JA | | | AVG | | |
|---|---|---|---|---|---|---|---|---|---|---|---|---|---|
| | | ACC | AUC | TPR@5 | ACC | AUC | TPR@5 | ACC | AUC | TPR@5 | ACC | AUC | TPR@5 |
| Llama3.2 -3B | Un-Water | 71.95 | - | - | 7.28 | - | - | 8.57 | - | - | 29.27 | - | - |
| | KGW | 64.06 | 0.963 | 0.814 | 6.29 | 0.982 | 0.900 | 6.44 | 0.985 | 0.920 | 25.60 | 0.980 | 0.878 |
| | Unigram | 61.64 | 0.938 | 0.724 | 6.44 | 0.960 | 0.828 | 6.75 | 0.970 | 0.860 | 24.94 | 0.960 | 0.804 |
| | UPV | 61.41 | 0.927 | 0.702 | 5.61 | 0.932 | 0.748 | 7.35 | 0.929 | 0.728 | 24.79 | 0.930 | 0.726 |
| | SWEET | 67.75 | 0.920 | 0.697 | 6.67 | 0.989 | 0.952 | 6.82 | 0.987 | 0.912 | 27.08 | 0.970 | 0.854 |
| | MorphMark | 72.25 | 0.750 | 0.253 | 7.43 | 0.745 | 0.272 | 9.17 | 0.786 | 0.368 | 29.62 | 0.760 | 0.298 |
| | KGW+D | 59.97 | **0.982** | **0.900** | 6.07 | **0.989** | **0.948** | **7.20** | **0.988** | **0.944** | 24.41 | **0.990** | **0.931** |
| | Unigram+D | 58.15 | **0.980** | **0.906** | 6.07 | **0.987** | **0.916** | 6.29 | **0.979** | **0.876** | 23.50 | **0.980** | **0.899** |
| | UPV+D | 57.09 | **0.977** | **0.888** | **5.84** | **0.967** | **0.848** | 5.76 | **0.968** | **0.864** | 22.90 | **0.970** | **0.867** |
| Qwen2.5 -3B | Un-Water | 33.06 | - | - | 9.25 | - | - | 7.51 | - | - | 16.61 | - | - |
| | KGW | 32.37 | 0.909 | 0.649 | 6.29 | 0.977 | 0.864 | 7.58 | 0.986 | 0.920 | 15.41 | 0.960 | 0.811 |
| | Unigram | 29.11 | 0.908 | 0.611 | 6.90 | 0.956 | 0.720 | 6.90 | 0.979 | 0.912 | 14.30 | 0.950 | 0.748 |
| | UPV | 29.11 | 0.856 | 0.477 | 5.91 | 0.958 | 0.756 | 5.91 | 0.934 | 0.736 | 13.64 | 0.920 | 0.715 |
| | SWEET | 34.57 | 0.898 | 0.590 | 6.82 | 0.994 | 0.972 | 7.51 | 0.990 | 0.948 | 16.30 | 0.960 | 0.837 |
| | MorphMark | 35.94 | 0.722 | 0.227 | 7.88 | 0.840 | 0.504 | 7.05 | 0.835 | 0.400 | 16.96 | 0.799 | 0.377 |
| | KGW+D | 31.01 | **0.973** | **0.883** | 6.60 | 0.974 | **0.872** | **7.96** | 0.972 | 0.880 | 15.19 | **0.970** | **0.878** |
| | Unigram+D | 27.14 | **0.963** | **0.822** | 7.66 | 0.918 | 0.572 | 6.90 | 0.955 | 0.768 | 13.90 | **0.950** | 0.721 |
| | UPV+D | 26.76 | **0.956** | **0.788** | 6.29 | 0.952 | 0.732 | 6.52 | 0.938 | 0.748 | 13.19 | **0.950** | **0.756** |
| Qwen3 -1.7B | Un-Water | 64.67 | - | - | 9.33 | - | - | 9.1 | - | - | 27.70 | - | - |
| | KGW | 62.55 | 0.696 | 0.198 | 8.64 | 0.780 | 0.284 | 8.57 | 0.774 | 0.336 | 26.59 | 0.750 | 0.273 |
| | Unigram | 63.08 | 0.711 | 0.208 | 8.64 | 0.749 | 0.232 | 8.26 | 0.829 | 0.388 | 26.66 | 0.760 | 0.276 |
| | UPV | 61.56 | 0.686 | 0.195 | 8.49 | 0.703 | 0.188 | 7.81 | 0.730 | 0.188 | 25.95 | 0.710 | 0.190 |
| | SWEET | 64.44 | 0.576 | 0.096 | 8.79 | 0.715 | 0.216 | 8.72 | 0.758 | 0.268 | 27.32 | 0.680 | 0.193 |
| | MorphMark | 65.35 | 0.576 | 0.086 | 8.87 | 0.604 | 0.136 | 8.42 | 0.644 | 0.108 | 27.55 | 0.608 | 0.110 |
| | KGW+D | 60.88 | **0.935** | **0.700** | 8.11 | **0.931** | **0.716** | **9.02** | **0.928** | **0.668** | 26.00 | **0.930** | **0.695** |
| | Unigram+D | **63.68** | **0.926** | **0.682** | 8.64 | **0.871** | **0.520** | 8.04 | **0.928** | **0.700** | **26.79** | **0.910** | **0.634** |
| | UPV+D | 56.33 | **0.902** | **0.644** | 7.35 | **0.845** | **0.384** | **8.19** | **0.867** | **0.472** | 23.96 | **0.870** | **0.500** |
| Qwen3 -4B | Un-Water | 55.42 | - | - | 11.68 | - | - | 10.39 | - | - | 25.83 | - | - |
| | KGW | 54.89 | 0.824 | 0.404 | 11.52 | 0.705 | 0.200 | 10.54 | 0.794 | 0.388 | 25.65 | 0.770 | 0.331 |
| | Unigram | 56.03 | 0.699 | 0.162 | 9.70 | 0.761 | 0.340 | 10.16 | 0.779 | 0.328 | 25.30 | 0.750 | 0.277 |
| | UPV | 56.41 | 0.669 | 0.175 | 11.37 | 0.639 | 0.156 | 9.78 | 0.743 | 0.276 | 25.85 | 0.680 | 0.202 |
| | SWEET | 47.46 | 0.694 | 0.202 | 11.60 | 0.685 | 0.184 | 10.39 | 0.718 | 0.260 | 23.15 | 0.700 | 0.215 |
| | MorphMark | 56.48 | 0.604 | 0.096 | 11.37 | 0.574 | 0.100 | 10.77 | 0.638 | 0.148 | 26.21 | 0.605 | 0.115 |
| | KGW+D | 53.90 | **0.974** | **0.861** | **11.68** | **0.904** | **0.560** | 9.70 | **0.941** | **0.708** | 25.09 | **0.940** | **0.710** |
| | Unigram+D | 55.88 | **0.916** | **0.631** | 10.84 | **0.857** | **0.508** | 9.86 | **0.902** | **0.652** | 25.53 | **0.890** | **0.597** |
| | UPV+D | 51.25 | **0.900** | **0.621** | 10.84 | **0.817** | **0.404** | 9.25 | **0.886** | **0.592** | 23.78 | **0.870** | **0.539** |

## C EXPERIMENTAL DETAILS

We evaluate on two additional datasets beyond PPL. For CNN/DailyMail (CNN/DM), we randomly sample 250 instances from the test dataset. For HumanEval, we use 164 coding problems.

## D LIMITATIONS

DynamicBias is training-free and detector-compatible. The remaining limits are pragmatic. (1) Theory uses a small-bias surrogate. Empirically, the trend persists beyond that regime. (2) We rely on top-k log-probabilities, which are widely available. When k is small or unavailable, we can use an entropy proxy. (3) We focus on common edits and paraphrases. Fully adaptive margin-shaping attacks are left for future work. (4) Very large models and niche domains are not exhaustively covered. Pilot checks show similar tendencies.

Table 11: Comparison to using an auxiliary model.

| OPT 1.3B | | | Llama2 13B | | |
|---|---|---|---|---|---|
| Method | PPL | AUC | Method | PPL | AUC |
| Un-Water | 12.32 | - | Un-Water | 6.16 | - |
| KGW | 22.84 | 0.998 | KGW | 12.30 | 0.997 |
| Unigram | 19.99 | **0.999** | Unigram | 12.16 | **0.998** |
| UPV | 11.56 | 0.994 | UPV | 8.16 | 0.993 |
| TS | 14.90 | 0.996 | TS | 7.62 | **0.998** |
| KGW + D | **13.48** | 0.994 | KGW + D | **7.57** | 0.997 |
| Unigram + D | **14.09** | 0.993 | Unigram + D | **7.58** | 0.994 |
| UPV + D | **11.41** | 0.965 | UPV + D | **6.43** | 0.957 |

Table 12: Effect of the pool size $c$ in the margin definition for each language.

| $c$ | C4 (EN) | | mC4 (KR) | | mC4 (JA) | | GSM (EN) | | MGSM (KR) | | MGSM (JA) | |
|---|---|---|---|---|---|---|---|---|---|---|---|---|
| | PPL↓ | AUC↑ | PPL↓ | AUC↑ | PPL↓ | AUC↑ | ACC↑ | AUC↑ | ACC↑ | AUC↑ | ACC↑ | AUC↑ |
| 2 | **4.79** | 0.993 | **4.88** | 0.982 | **5.62** | 0.975 | 49.73 | 0.919 | 11.52 | 0.841 | **9.86** | 0.889 |
| 5 | 5.38 | 0.998 | 5.59 | 0.994 | 6.39 | 0.990 | **53.90** | 0.974 | **11.68** | 0.904 | 9.70 | 0.941 |
| 10 | 5.99 | **0.999** | 6.37 | **0.995** | 7.19 | **0.993** | **53.90** | **0.990** | 10.92 | **0.960** | 9.55 | **0.975** |

Table 13: Performance of sequence-aware calibration and token-level DynamicBias for each language.

| Scope | C4 (EN) | | mC4 (KR) | | mC4 (JA) | | GSM (EN) | | MGSM (KR) | | MGSM (JA) | |
|---|---|---|---|---|---|---|---|---|---|---|---|---|
| | PPL | AUC | PPL | AUC | PPL | AUC | ACC | AUC | ACC | AUC | ACC | AUC |
| Morphmark ($k_{linear} = 1.55$) | **4.5** | 0.8951 | **4.33** | 0.8980 | 4.88 | 0.8654 | 56.48 | 0.6042 | 11.37 | 0.5742 | **10.77** | 0.6375 |
| Morphmark ($k_{linear} = 9.0$) | 5.03 | 0.9935 | 5.44 | 0.9889 | 5.79 | 0.9748 | 56.25 | 0.7357 | **11.75** | 0.7237 | 10.16 | 0.7802 |
| Token | 4.69 | 0.9487 | 5.02 | 0.8646 | **4.55** | 0.8692 | **57.7** | 0.7944 | 11.68 | 0.6442 | 10.46 | 0.7096 |
| Sequence | 5.38 | **0.9983** | 5.59 | **0.9942** | 6.39 | **0.9903** | 53.9 | **0.9736** | 11.68 | **0.9037** | 9.7 | **0.9414** |

Table 14: Average number of green tokens across dataset.

| Method | C4 (Green / Total) | | | GSM8K (Green / Total) | | |
|---|---|---|---|---|---|---|
| | Original | WordDel | WordSub | Original | WordDel | WordSub |
| KGW | 98.58 / 198.36 | 39.57 / 97.90 | 87.13 / 252.72 | 52.67 / 181.27 | 21.56 / 85.10 | 53.90 / 191.12 |
| KGW+D | **100.93** / 198.67 | **40.07** / 98.14 | **87.92** / 253.06 | **66.14** / 182.61 | **24.75** / 85.89 | **57.19** / 192.78 |

Table 15: Performance of each method on C4 and mC4 across languages with top-$p$ sampling.

| Model | Method | EN | | | KR | | | JA | | | AVG | | |
|---|---|---|---|---|---|---|---|---|---|---|---|---|---|
| | | PPL | AUC | TPR@5 | PPL | AUC | TPR@5 | PPL | AUC | TPR@5 | PPL | AUC | TPR@5 |
| Llama3.2 -3B | Un-Water | 3.79 | - | - | 5.09 | - | - | 6.23 | - | - | 5.04 | - | - |
| | KGW | 6.77 | 0.997 | 0.992 | 11.52 | 0.989 | 0.976 | 11.53 | 0.995 | 0.976 | 9.94 | 0.994 | 0.982 |
| | Unigram | 7.64 | 0.998 | 0.996 | 7.00 | 0.995 | 0.978 | 10.27 | 0.996 | 0.992 | 8.30 | 0.997 | 0.989 |
| | UPV | 5.92 | 1.000 | 0.996 | 7.14 | 0.975 | 0.902 | 8.38 | 0.975 | 0.920 | 7.15 | 0.983 | 0.939 |
| | SWEET | 5.94 | 0.996 | 0.990 | 11.06 | 0.989 | 0.972 | 11.11 | 0.992 | 0.982 | 9.37 | 0.992 | 0.981 |
| | MorphMark | 3.90 | 0.977 | 0.886 | 5.99 | 0.905 | 0.716 | 6.40 | 0.916 | 0.680 | 5.43 | 0.932 | 0.761 |
| | KGW+D | **5.05** | **1.000** | **0.998** | **8.87** | **0.993** | **0.982** | **9.28** | 0.994 | **0.984** | **7.73** | **0.996** | **0.988** |
| | Unigram+D | **5.21** | **0.999** | **0.996** | 8.04 | 0.992 | 0.968 | **9.81** | 0.989 | 0.976 | **7.69** | 0.994 | 0.980 |
| | UPV+D | **4.92** | 0.999 | 0.988 | 7.41 | 0.945 | 0.776 | **7.87** | 0.961 | 0.864 | **6.73** | 0.968 | 0.876 |
| Qwen2.5 -3B | Un-Water | 4.75 | - | - | 3.77 | - | - | 3.81 | - | - | 4.11 | - | - |
| | KGW | 10.57 | 0.993 | 0.984 | 8.57 | 0.974 | 0.942 | 8.79 | 0.989 | 0.982 | 9.31 | 0.985 | 0.969 |
| | Unigram | 9.45 | 0.989 | 0.964 | 7.99 | 0.967 | 0.916 | 8.70 | 0.984 | 0.970 | 8.71 | 0.980 | 0.950 |
| | UPV | 9.26 | 0.993 | 0.980 | 6.53 | 0.947 | 0.886 | 6.60 | 0.982 | 0.960 | 7.46 | 0.974 | 0.942 |
| | SWEET | 9.57 | 0.989 | 0.972 | 7.47 | 0.950 | 0.908 | 7.66 | 0.980 | 0.958 | 8.23 | 0.973 | 0.946 |
| | MorphMark | 5.01 | 0.967 | 0.950 | 3.96 | 0.896 | 0.736 | 4.04 | 0.923 | 0.762 | 4.34 | 0.929 | 0.816 |
| | KGW+D | **5.85** | **1.000** | **1.000** | **5.81** | **0.994** | **0.980** | **6.07** | **0.995** | **0.984** | **5.91** | **0.996** | **0.988** |
| | Unigram+D | **5.74** | **0.994** | **0.982** | **5.89** | **0.991** | **0.980** | **6.19** | **0.988** | **0.974** | **5.94** | **0.991** | **0.979** |
| | UPV+D | **5.98** | **0.996** | **0.992** | **5.31** | **0.977** | **0.936** | **5.50** | 0.981 | 0.942 | **5.60** | **0.985** | **0.957** |
| Qwen3 -1.7B | Un-Water | 5.23 | - | - | 3.95 | - | - | 4.17 | - | - | 4.45 | - | - |
| | KGW | 6.17 | 0.979 | 0.950 | 5.50 | 0.908 | 0.824 | 6.04 | 0.936 | 0.884 | 5.90 | 0.941 | 0.886 |
| | Unigram | 6.15 | 0.977 | 0.942 | 5.02 | 0.942 | 0.870 | 5.51 | 0.947 | 0.806 | 5.56 | 0.955 | 0.873 |
| | UPV | 6.21 | 0.985 | 0.948 | 4.96 | 0.889 | 0.784 | 5.35 | 0.908 | 0.788 | 5.51 | 0.928 | 0.840 |
| | SWEET | 5.88 | 0.979 | 0.942 | 5.05 | 0.910 | 0.820 | 5.42 | 0.919 | 0.840 | 5.45 | 0.936 | 0.867 |
| | MorphMark | 5.28 | 0.896 | 0.542 | 3.96 | 0.840 | 0.462 | 4.15 | 0.803 | 0.482 | 4.46 | 0.847 | 0.495 |
| | KGW+D | 6.38 | **0.990** | **0.976** | 5.46 | **0.960** | **0.910** | 6.23 | **0.970** | **0.944** | 6.02 | **0.973** | **0.943** |
| | Unigram+D | 6.18 | **0.978** | **0.942** | 5.13 | **0.961** | **0.876** | 5.78 | **0.968** | **0.894** | 5.70 | **0.969** | **0.904** |
| | UPV+D | 6.32 | **0.991** | **0.958** | 5.12 | **0.940** | **0.858** | 5.77 | **0.940** | **0.840** | 5.74 | **0.957** | **0.885** |
| Qwen3 -4B | Un-Water | 4.54 | - | - | 4.41 | - | - | 4.96 | - | - | 4.64 | - | - |
| | KGW | 5.39 | 0.998 | 0.998 | 5.96 | 0.984 | 0.958 | 6.57 | 0.981 | 0.952 | 5.97 | 0.988 | 0.969 |
| | Unigram | 5.43 | 0.996 | 0.992 | 5.57 | 0.984 | 0.928 | 5.99 | 0.963 | 0.824 | 5.66 | 0.981 | 0.915 |
| | UPV | 5.30 | 0.992 | 0.990 | 5.44 | 0.953 | 0.878 | 5.96 | 0.933 | 0.822 | 5.57 | 0.959 | 0.897 |
| | SWEET | 4.97 | 0.996 | 0.982 | 5.48 | 0.988 | 0.966 | 6.00 | 0.980 | 0.936 | 5.48 | 0.988 | 0.961 |
| | MorphMark | 4.59 | 0.894 | 0.518 | 4.46 | 0.886 | 0.528 | 4.97 | 0.868 | 0.450 | 4.67 | 0.882 | 0.499 |
| | KGW+D | 5.44 | **0.998** | 0.996 | **5.79** | **0.994** | **0.984** | 6.54 | **0.988** | **0.980** | 5.92 | **0.993** | **0.987** |
| | Unigram+D | 5.46 | **0.997** | **0.996** | 5.62 | **0.987** | **0.946** | 6.35 | **0.972** | **0.862** | 5.81 | **0.985** | **0.935** |
| | UPV+D | 5.44 | **0.994** | **0.994** | 5.53 | **0.973** | **0.926** | 6.02 | **0.954** | **0.856** | 5.66 | **0.974** | **0.925** |

Table 16: Performance of each method on GSM and MGSM across languages with top-$p$ sampling.

| Model | Method | EN | | | KR | | | JA | | | AVG | | |
|-------|--------|-----|-----|-------|-----|-----|-------|-----|-----|-------|-----|-----|-------|
| | | ACC | AUC | TPR@5 | ACC | AUC | TPR@5 | ACC | AUC | TPR@5 | ACC | AUC | TPR@5 |
| Llama3.2 -3B | Un-Water | 72.02 | - | - | 7.66 | - | - | 8.49 | - | - | 29.39 | - | - |
| | KGW | 61.71 | 0.961 | 0.817 | 6.22 | 0.984 | 0.924 | 6.14 | 0.985 | 0.908 | 24.69 | 0.977 | 0.883 |
| | Unigram | 60.12 | 0.939 | 0.718 | 5.69 | 0.975 | 0.872 | 6.14 | 0.975 | 0.868 | 23.98 | 0.963 | 0.819 |
| | UPV | 58.45 | 0.929 | 0.707 | 5.91 | 0.946 | 0.776 | 6.52 | 0.944 | 0.788 | 23.63 | 0.940 | 0.757 |
| | SWEET | 68.54 | 0.915 | 0.683 | 6.60 | 0.992 | 0.968 | 6.90 | 0.988 | 0.936 | 27.35 | 0.965 | 0.862 |
| | MorphMark | 70.96 | 0.742 | 0.263 | 7.28 | 0.740 | 0.240 | 9.10 | 0.768 | 0.312 | 29.11 | 0.750 | 0.272 |
| | KGW+D | 60.35 | **0.980** | **0.890** | 5.61 | **0.992** | **0.968** | 6.37 | **0.994** | **0.984** | 24.11 | **0.989** | **0.947** |
| | Unigram+D | 57.16 | **0.982** | **0.917** | 5.91 | **0.991** | **0.960** | 5.99 | **0.983** | **0.908** | 23.02 | **0.985** | **0.928** |
| | UPV+D | 54.44 | **0.968** | **0.837** | 6.22 | **0.954** | **0.776** | 6.75 | **0.956** | **0.808** | 22.47 | **0.959** | **0.807** |
| Qwen2.5 -3B | Un-Water | 34.12 | - | - | 8.79 | - | - | 7.58 | - | - | 16.83 | - | - |
| | KGW | 31.54 | 0.913 | 0.646 | 6.75 | 0.978 | 0.876 | 7.43 | 0.986 | 0.944 | 15.24 | 0.959 | 0.822 |
| | Unigram | 28.73 | 0.902 | 0.597 | 6.52 | 0.958 | 0.776 | 5.69 | 0.982 | 0.920 | 13.65 | 0.947 | 0.764 |
| | UPV | 29.26 | 0.856 | 0.487 | 5.46 | 0.959 | 0.780 | 6.52 | 0.943 | 0.776 | 13.75 | 0.919 | 0.681 |
| | SWEET | 33.21 | 0.895 | 0.604 | 7.20 | 0.993 | 0.952 | 6.97 | 0.991 | 0.944 | 15.79 | 0.960 | 0.833 |
| | MorphMark | 36.01 | 0.732 | 0.233 | 7.28 | 0.819 | 0.444 | 6.97 | 0.836 | 0.400 | 16.75 | 0.796 | 0.359 |
| | KGW+D | 30.48 | **0.976** | **0.887** | 6.67 | **0.982** | **0.876** | **7.73** | 0.978 | 0.904 | 14.96 | **0.979** | **0.889** |
| | Unigram+D | 26.61 | **0.969** | **0.849** | **6.90** | 0.917 | 0.576 | **6.75** | 0.969 | 0.856 | 13.42 | **0.952** | 0.760 |
| | UPV+D | 27.22 | **0.937** | **0.709** | **6.07** | 0.937 | 0.688 | 5.99 | 0.931 | 0.712 | 13.09 | **0.935** | **0.703** |
| Qwen3 -1.7B | Un-Water | 64.29 | - | - | 9.40 | - | - | 8.57 | - | - | 27.42 | - | - |
| | KGW | 62.62 | 0.699 | 0.204 | 9.10 | 0.783 | 0.320 | 8.72 | 0.781 | 0.352 | 26.81 | 0.754 | 0.292 |
| | Unigram | 63.23 | 0.711 | 0.211 | 8.72 | 0.741 | 0.260 | 8.34 | 0.843 | 0.420 | 26.76 | 0.765 | 0.297 |
| | UPV | 62.02 | 0.684 | 0.197 | 7.96 | 0.704 | 0.176 | 7.81 | 0.735 | 0.212 | 25.93 | 0.708 | 0.195 |
| | SWEET | 64.06 | 0.576 | 0.092 | 8.95 | 0.716 | 0.208 | 8.57 | 0.755 | 0.272 | 27.19 | 0.682 | 0.191 |
| | MorphMark | 64.22 | 0.576 | 0.090 | 8.87 | 0.600 | 0.120 | 8.64 | 0.642 | 0.100 | 27.24 | 0.606 | 0.103 |
| | KGW+D | 60.50 | **0.936** | **0.697** | 8.72 | **0.929** | **0.700** | 9.02 | **0.930** | **0.688** | 26.08 | **0.931** | **0.695** |
| | Unigram+D | **63.23** | 0.926 | 0.676 | 9.17 | 0.874 | 0.508 | 7.96 | 0.926 | 0.656 | **26.79** | 0.909 | 0.613 |
| | UPV+D | 57.01 | **0.872** | **0.558** | 7.13 | **0.823** | 0.352 | 7.96 | 0.842 | 0.416 | 24.03 | **0.846** | **0.479** |
| Qwen3 -4B | Un-Water | 55.27 | - | - | 11.52 | - | - | 10.46 | - | - | 25.75 | - | - |
| | KGW | 54.51 | 0.824 | 0.404 | 11.45 | 0.710 | 0.200 | 10.24 | 0.799 | 0.396 | 25.40 | 0.777 | 0.333 |
| | Unigram | 56.41 | 0.701 | 0.166 | 9.55 | 0.759 | 0.332 | 10.31 | 0.790 | 0.340 | 25.42 | 0.750 | 0.279 |
| | UPV | 56.18 | 0.669 | 0.169 | 11.98 | 0.644 | 0.140 | 9.78 | 0.743 | 0.276 | 25.98 | 0.685 | 0.195 |
| | SWEET | 48.07 | 0.689 | 0.188 | 11.68 | 0.666 | 0.192 | 10.24 | 0.719 | 0.284 | 23.33 | 0.691 | 0.221 |
| | MorphMark | 56.56 | 0.608 | 0.103 | 11.30 | 0.586 | 0.100 | 10.61 | 0.627 | 0.120 | 26.16 | 0.607 | 0.108 |
| | KGW+D | 53.98 | **0.973** | **0.861** | **11.98** | **0.903** | **0.552** | 9.78 | **0.948** | **0.744** | 25.25 | **0.941** | **0.719** |
| | Unigram+D | 55.34 | **0.919** | **0.626** | 10.99 | **0.863** | **0.520** | 10.08 | **0.905** | **0.648** | **25.47** | **0.895** | **0.598** |
| | UPV+D | 53.30 | **0.871** | **0.550** | 10.54 | **0.780** | **0.308** | 10.01 | **0.870** | **0.572** | 24.62 | **0.840** | **0.477** |

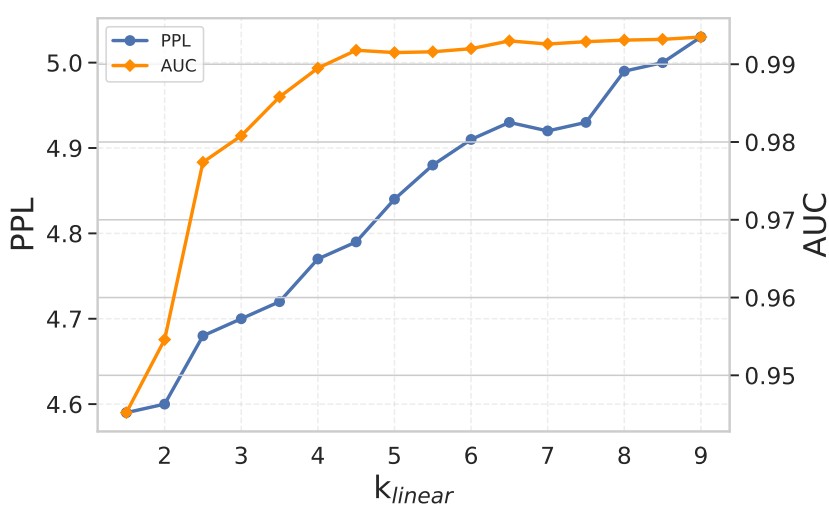

Figure 6: Increasing $k_{linear}$ parameter for MorphMark.

