# OpenReview forum: "DynamicBias: Sequence-Aware Calibrated Watermarking for Large Language Models"
_ICLR.cc/2026/Conference — Submitted to ICLR 2026_

### Official Review · Reviewer_L388 · 2025-10-31

**Soundness:** 4
**Presentation:** 3
**Contribution:** 3
**Rating:** 6
**Confidence:** 4

**Summary:**

This paper proposes an enhancement for green/red list text watermarking methods. Instead of using a fixed bias ($\delta$), this method calculates the bias dynamically at each generation step.It is set as a function of the sequence-level average of the model's certainty margin which is the gap between the top logit and the average of the next c logits. This makes the watermark more stable and effective across different models, domains, and languages as demonstrated in thorough experiments.

**Strengths:**

S1. Clear motivation and writing The paper is clear and well-written. It provides a strong motivation for its approach by demonstrating, in a preliminary study (Section 4.2), that static bias methods have inconsistent performance due to significant variation in logit distributions across models and domains (e.g., C4 vs. GSM8K).

S2. Thorough evaluation. The experimental evaluation is comprehensive and rigorous. The method is tested across: multiple models (Llama, Qwen), diverse tasks (open-ended generation, math reasoning, code generation), multiple languages (English, Japanese, Korean).

S3. Ablation studies. The paper includes excellent ablation studies analyzing the effect of every key hyperparameter, including: scaling factor \alpha, green list proportion \gamma, pool size c, the impact of the sequence-level averaging itself

S4. Theoretical justification. The proposed method is supported by a theoretical analysis (Section 3.4), which shows that dynamic calibration maximizes the expected detection statistic under a quadratic cost budget. (I havent had the time to look at the proofs though).

**Weaknesses:**

W1. The method shows limited conceptual novelty, as it is quite similar to Morphmark, which also employed a token-level dynamic bias.
The main innovations of DynamicBias appear to be: the use of a sequence-level average ($\overline{m}_t$) rather than an instantaneous signal, the use of a logit margin rather than probability mass.

**Questions:**

Q1. The core novelty lies in the use of a sequence-level average $\overline{m}_t$. How is this average handled in practice?
Is it reset for every new prompt?
If not, then a very long, high-certainty sequence (e.g., code generation) followed by a short, low-certainty sequence (e.g., poetry) could result in an improperly high bias for the second sequence.
If it is reset, how does the method perform at the beginning of a sequence, when $\overline{m}_t$ is based on only one or two tokens and thus not yet a stable “sequence-level” average?

---

> ### Author Response · Authors · 2025-11-21
>
> Hi! We sincerely appreciate your careful reading and feedback to improve our draft.
>
> > W1. The method shows limited conceptual novelty, as it is quite similar to Morphmark, which also employed a token-level dynamic bias. The main innovations of DynamicBias appear to be: the use of a sequence-level average ($\overline{m_t}$) rather than an instantaneous signal, the use of a logit margin rather than probability mass.
>
> We appreciate the opportunity to clarify the differences. A key practical diction is at detection time. SWEET requires explicitly querying the model to obtain token-level probabilities (or entropies) for each generated sequence, which incurs substantial computational overhead during detection. In contrast, DynamicBias is designed to remain fully compatible with the standard KGW detector and does not require any additional model calls or entropy computation at detection time.
>
> Conceptually, prior adaptive methods operate on token-level entropy or probabilities, whereas DynamicBias uses the sequence-averaged logit margin as its control signal and applies a single global parameter α, which we analyze under a small-bias regime and find to work consistently across different models, domains, and languages. While we do not claim this to be groundbreaking, we believe that the  model-, domain-, and language-agnostic nature of DynamicBias provides a meaningful and practically relevant contribution beyond existing adaptive watermarking schemes.

---

> ### Author Response · Authors · 2025-11-21
>
> > Q1. The core novelty lies in the use of a sequence-level average $\overline{m_t}$. How is this average handled in practice? Is it reset for every new prompt? If not, then a very long, high-certainty sequence (e.g., code generation) followed by a short, low-certainty sequence (e.g., poetry) could result in an improperly high bias for the second sequence. If it is reset, how does the method perform at the beginning of a sequence, when $\overline{m_t}$ is based on only one or two tokens and thus not yet a stable “sequence-level” average?
>
> We appreciate your insightful question. We reset the sequence-level average  for every new prompt to avoid cross-sequence contamination. We acknowledge that, at the very beginning of a sequence, an average based on only one or two tokens may not yet be fully stable. To investigate this, we analyzed detection performance as a function of the generated sequence length on the C4 dataset using the Qwen3-4B. Specifically, we first generated all sequences and then grouped them into bins based on the number of generated tokens, measuring detection performance within each bin. We observe that DynamicBias does not overly dampen the bias, instead it stabilizes the z-scores and improves detectability. We will include this in the draft.
>
>
>
> | Data    | Method | 0–25 AUC | 0–25 TPR@5 | 25–50 AUC | 25–50 TPR@5 | 50–75 AUC | 50–75 TPR@5 | 75–100 AUC | 75–100 TPR@5 |
> |---------|--------|----------|-------------|------------|--------------|------------|--------------|-------------|----------------|
> | C4-EN   | KGW    | 0.9262   | 0.706       | 0.9755     | 0.886        | 0.9914     | 0.940        | 0.9943      | 0.978          |
> |         | KGW+D  | **0.9500** | **0.790**   | **0.9835** | **0.920**    | **0.9929** | **0.966**    | **0.9952**  | **0.986**      |
> |---------|--------|----------|-------------|------------|--------------|------------|--------------|-------------|----------------|
> | C4-KR   | KGW    | 0.8935   | 0.586       | 0.9423     | 0.754        | 0.9626     | 0.888        | 0.9717      | 0.910          |
> |         | KGW+D  | **0.9249** | **0.676**   | **0.9591** | **0.820**    | **0.9777** | **0.912**    | **0.9857**  | **0.950**      |
> |---------|--------|----------|-------------|------------|--------------|------------|--------------|-------------|----------------|
> | C4-JA   | KGW    | 0.8520   | 0.500       | 0.9185     | 0.782        | 0.9499     | 0.840        | 0.9588      | 0.888          |
> |         | KGW+D  | **0.8868** | **0.586**   | **0.9468** | **0.836**    | **0.9674** | **0.900**    | **0.9805**  | **0.946**      |

---

### Official Review · Reviewer_jV92 · 2025-11-01

**Soundness:** 4
**Presentation:** 4
**Contribution:** 3
**Rating:** 8
**Confidence:** 4

**Summary:**

This paper introduces DynamicBias, a sequence-aware watermarking framework that adapts the watermarking bias dynamically during generation based on model certainty. Building on the standard KGW green/red list formulation, DynamicBias replaces the static bias δ with a stepwise bias δₜ = α m̄ₜ, where m̄ₜ is a sequence-level average of the logit margin between the top token and its next-best competitors. This sequence-level calibration adjusts watermark strength to the model’s confidence, stronger when the model is certain, weaker when uncertain, yielding more consistent detectability and text quality across models, domains, and languages. Extensive experiments across four LLMs and three languages (English, Japanese, Korean) on C4/mC4, GSM8K/MGSM, and additional summarization and code generation benchmarks demonstrate robust performance gains in both AUC and perplexity, with improved paraphrase resilience.

**Strengths:**

* The idea of calibrating watermark bias according to sequence-level model confidence is intuitive and effective. By maintaining a running certainty signal, the approach captures longer-range structure missed by token-level entropy gating.
* The experiments are comprehensive, spanning multiple models, languages, and task types (reasoning, summarization, code), and consistently show detectability and robustness improvements with minimal quality loss.
* Compared to token-level entropy- or probability-mass–based scaling, DynamicBias’s use of sequence-averaged logits introduces a smoother and more stable watermarking dynamic that generalizes better across domains.
* The authors validate the theoretical small-bias regime assumptions empirically through linearity and curvature fits, and ablate key hyperparameters (α, γ, c), adding strong credibility and tangible implementation recommendations for practitioners.

**Weaknesses:**

* The approach is conceptually close to recent entropy- and context-aware watermarking schemes, which makes the distinction of maintaining a sequence-level margin rather than a token-level entropy somewhat incremental. That said, the theoretical framing is elegant, and the study is executed well.
* The paper does not report generation throughput or wall-clock time relative to vanilla watermarking. Even if the per-step overhead is small, empirical confirmation would strengthen claims of practical deployability.
* While results are strong overall, the paper could better quantify trade-offs—e.g., where DynamicBias might underperform in high-entropy tasks or non-English domains (given these were tested).

**Questions:**

* Could the authors provide runtime comparisons or throughput (tokens/s) relative to static watermarking?
* Are there scenarios (e.g., very short or noisy generations) where sequence-level averaging may dampen the bias too much, hurting detectability?
* When the model’s confidence shifts sharply within a sequence (e.g., from setup to reasoning to conclusion, or natural language/explanation to code), does $\delta_t$ adapt smoothly, or could lag in the running average cause miscalibration? A short visualization of $\delta_t$ evolution over token steps could clarify this.
* Since $\delta_t$ varies across sequences, is any adjustment to the detector threshold or z-score normalization needed to maintain comparable false-positive rates across tasks?

---

> ### Author Response · Authors · 2025-11-21
>
> Hi! We sincerely appreciate your careful reading and feedback to improve our draft.
>
> > W1. The approach is conceptually close to recent entropy- and context-aware watermarking schemes, which makes the distinction of maintaining a sequence-level margin rather than a token-level entropy somewhat incremental. That said, the theoretical framing is elegant, and the study is executed well.
>
> We appreciate the opportunity to clarify the differences. A key practical diction is at detection time. SWEET requires explicitly querying the model to obtain token-level probabilities (or entropies) for each generated sequence, which incurs substantial computational overhead during detection. In contrast, DynamicBias is designed to remain fully compatible with the standard KGW detector and does not require any additional model calls or entropy computation at detection time.
>
> Conceptually, prior adaptive methods operate on token-level entropy or probabilities, whereas DynamicBias uses the sequence-averaged logit margin as its control signal and applies a single global parameter α, which we analyze under a small-bias regime and find to work consistently across different models, domains, and languages. While we do not claim this to be groundbreaking, we believe that the  model-, domain-, and language-agnostic nature of DynamicBias provides a meaningful and practically relevant contribution beyond existing adaptive watermarking schemes.
>
> ---
>
> > W2. The paper does not report generation throughput or wall-clock time relative to vanilla watermarking. Even if the per-step overhead is small, empirical confirmation would strengthen claims of practical deployability.
>
> We appreciate your careful reading and constructive feedback. Following your feedback, we conducted additional experiments using three models of different sizes. To evaluate runtime, we sampled 100 instances from the C4 dataset, generated 200 tokens for each instance, and computed the average latency on an NVIDIA A6000 GPU.
>
> The results show that Dynamic Bias does not significantly increase generation time compared to the baseline methods. At detection time, DynamicBias incurs almost no additional overhead because it does not require computing token entropies or querying the model, unlike some entropy-based methods. We will incorporate these results and a detailed description of the setup into the revised draft.
>
> | Model          | Method     | Generation Time (token/ms) | Generation Time (data/s) | Detection Time (data/ms) |
> |----------------|------------|-----------------------------|----------------------------|----------------------------|
> | **Llama 3.2 1B** | KGW        | 11.75 | 2.30 | 25.1 |
> |                | KGW + D   | 11.90 | 2.31 | 25.0 |
> |                | UPV        | 14.23 | 2.78 | 41.5 |
> |                | UPV + D   | 14.38 | 2.78 | 41.5 |
> |                | SWEET      | 11.88 | 2.30 | 68.4 |
> |                | Morphmark  | 12.14 | 2.33 | 26.1 |
> |----------------|------------|-----------------------------|----------------------------|----------------------------|
> | **Llama 3.2 3B** | KGW        | 26.28 | 5.25 | 24.9 |
> |                | KGW + D   | 26.41 | 5.28 | 25.0 |
> |                | UPV        | 28.61 | 5.71 | 38.5 |
> |                | UPV + D   | 28.74 | 5.74 | 38.9 |
> |                | SWEET      | 26.37 | 5.27 | 133.6 |
> |                | Morphmark  | 26.64 | 5.28 | 26.8 |
> |----------------|------------|-----------------------------|----------------------------|----------------------------|
> | **Qwen 3 4B**  | KGW        | 34.93 | 6.97 | 26.2 |
> |                | KGW + D   | 35.17 | 7.03 | 26.3 |
> |                | UPV        | 37.92 | 7.57 | 44.1 |
> |                | UPV + D   | 38.03 | 7.61 | 44.6 |
> |                | SWEET      | 35.18 | 7.01 | 163.4 |
> |                | Morphmark  | 35.54 | 7.06 | 28.1 |
>
> ---
> > W3. While results are strong overall, the paper could better quantify trade-offs—e.g., where DynamicBias might underperform in high-entropy tasks or non-English domains (given these were tested).
>
> We agree that the trade-offs more explicit would strengthen the paper. We believe the remaining gaps mainly stem from differences in logit gaps across tasks and languages (Figure 2). While our DynamicBias controls the signal via sequence-averaged margin, we currently use a single fixed &alpha; across all settings, which can yield suboptimal trade-offs in some regimes. We will clarify this limitation in the revision and, as noted in the conclusion, plan to explore learning alpha per task or domain the future work.

---

> ### Author Response · Authors · 2025-11-21
>
> > Q1. Could the authors provide runtime comparisons or throughput (tokens/s) relative to static watermarking?
>
> Following your feedback, we have included the results in the above response (W2.). Once again, thank you for your thoughtful comments.
>
> > Q2. Are there scenarios (e.g., very short or noisy generations) where sequence-level averaging may dampen the bias too much, hurting detectability?
>
> We appreciate your valuable feedback. To address this concern, we analyzed detection performance as a function of the generated sequence length on the C4 dataset using the Qwen3-4B. Specifically, we first generated all sequences and then grouped them into bins based on the number of generated tokens, measuring detection performance within each bin. We observe that DynamicBias does not overly dampen the bias, instead it stabilizes the z-scores and improves detectability. We will include this in the draft.
>
>
> | Data    | Method | 0–25 AUC | 0–25 TPR@5 | 25–50 AUC | 25–50 TPR@5 | 50–75 AUC | 50–75 TPR@5 | 75–100 AUC | 75–100 TPR@5 |
> |---------|--------|----------|-------------|------------|--------------|------------|--------------|-------------|----------------|
> | C4-EN   | KGW    | 0.9262   | 0.706       | 0.9755     | 0.886        | 0.9914     | 0.940        | 0.9943      | 0.978          |
> |         | KGW+D  | **0.9500** | **0.790**   | **0.9835** | **0.920**    | **0.9929** | **0.966**    | **0.9952**  | **0.986**      |
> |---------|--------|----------|-------------|------------|--------------|------------|--------------|-------------|----------------|
> | C4-KR   | KGW    | 0.8935   | 0.586       | 0.9423     | 0.754        | 0.9626     | 0.888        | 0.9717      | 0.910          |
> |         | KGW+D  | **0.9249** | **0.676**   | **0.9591** | **0.820**    | **0.9777** | **0.912**    | **0.9857**  | **0.950**      |
> |---------|--------|----------|-------------|------------|--------------|------------|--------------|-------------|----------------|
> | C4-JA   | KGW    | 0.8520   | 0.500       | 0.9185     | 0.782        | 0.9499     | 0.840        | 0.9588      | 0.888          |
> |         | KGW+D  | **0.8868** | **0.586**   | **0.9468** | **0.836**    | **0.9674** | **0.900**    | **0.9805**  | **0.946**      |
>
> ---
>
> > Q3. When the model’s confidence shifts sharply within a sequence (e.g., from setup to reasoning to conclusion, or natural language/explanation to code), does $\delta_{t}$ adapt smoothly, or could lag in the running average cause miscalibration? A short visualization of $\delta_{t}$ evolution over token steps could clarify this.
>
> We appreciate this question. In dynamicBias, we set $\delta_{t}=\alpha*\overline{m_t}$ and update it as a running average of the token-wise margins, so $\delta_{t}$ is updated at every step and changes smoothly as the model confidence shifts. This introduces some inertia by design, but with a conservative scale &alpha;, we did not observe any obvious miscalibration spikes. Following your suggestion, we will add a simple plot of $\overline{m_t}$ and $\delta_{t}$ over token positions for representative sequences to visualize how $\delta_{t}$ tracks different phases within a generation.
>
> ---
>
> > Q4. Since $\delta_{t}$ varies across sequences, is any adjustment to the detector threshold or z-score normalization needed to maintain comparable false-positive rates across tasks?
>
> We appreciate this question. In our implementation, we do not modify the standard KGW z-score computation or the decision threshold. DynamicBias only changes $\delta_{t}$ during generation while keeping the hash partition and z-statistic identical to KGW, and unwatermarked text corresponds to $\delta_{t}=0$. As a result, the null distribution of the z-score ( and thus the false-positive rate for a fixed threshold) remains essentially unchanged. Empirically, we observe very similar FPR across tasks and languages for DynamicBias and the static KGW baseline when using the same threshold (and our main results are reported in terms of AUC which is threshold-free). We will clarify this point in the revised draft.

---

### Official Review · Reviewer_5nZ9 · 2025-11-03

**Soundness:** 2
**Presentation:** 3
**Contribution:** 3
**Rating:** 4
**Confidence:** 3

**Summary:**

The paper proposed a method to dynamically update the bias $\delta$ in green-red list type watermarking methods. The paper proposes to quantify stepwise certainty using a margin between the top logit and a small pool of next-best logits, and use a sequence-level average of the margin over the generated steps to update the bias $\delta$ at each step. Empirical experiments show that by incorporating the proposed method in updating $\delta$, existing green-red list watermarking methods can improve detectability and generation quality.

**Strengths:**

The paper clearly explained the existing issue of using a static $\delta$ value in the green-red list type watermarks. The proposed method is clearly motivated. Empirical results demonstrate that introducing the dynamic setting of the $\delta$ value in the existing watermarking method can improve the detectability and text quality

**Weaknesses:**

The paper conducts experiments across different LLMs and datasets, but there are some related baseline missing and some comparison are not well discussed. Specifically:

1. The proposed method dynamically sets $\delta$ value, there are related baselines, which also dynamically update $\delta$ values,  are not compared. For instance, I think the method in [1], which learns an auxiliary model to predict $\delta, \gamma$ at each step, can serve as a natural baseline. It is briefly discussed in line 59, I agree that the method may not be model-independent, but they do have some pre-trained auxiliary models for the Llama and OPT family in the GitHub repo, so I think a comparison in some families should be doable.

2. In the comparison with MorphMark, the paper states that "MorphMark shows weaker detectability overall". But MorphMark also consistently shows better text quality, e.g., perplexity and acc in Table 1. Is it possible to tune the MorphMark model to achieve similar detectability, then compare text quality, or achieve similar text quality, then compare detectability?


### References

[1] Huo, Mingjia, et al. "Token-specific watermarking with enhanced detectability and semantic coherence for large language models." arXiv preprint arXiv:2402.18059 (2024).

**Questions:**

Please refer to weakness

---

> ### Author Response · Authors · 2025-11-21
>
> Hi! We sincerely appreciate your careful reading and feedback to improve our draft.
>
> > W1. The proposed method dynamically sets &delta; value, there are related baselines, which also dynamically update &delta; values, are not compared. For instance, I think the method in [1], which learns an auxiliary model to predict &delta;, &gamma; at each step, can serve as a natural baseline. It is briefly discussed in line 59, I agree that the method may not be model-independent, but they do have some pre-trained auxiliary models for the Llama and OPT family in the GitHub repo, so I think a comparison in some families should be doable.
>
> We appreciate your valuable feedback. As you noted,[1] is not model-independent. To compare it fairly with our method, we evaluated it on both OPT and LLama models on the C4-english dataset. The results are as follows. We observed that DynamicBias outperforms both using OPT and Llama models. The results were conducted following the TS watermarking method’s configuration, with Temperature=1 and Top-k=50.
>
> | Model     | Method         | PPL   | AUC   |
> |---------|---------------|-----|-----|
> | OPT 1.3B  | Wo Watermark   | 12.32 | -     |
> |           | KGW            | 22.84 | 0.998 |
> |           | Unigram        | 19.99 | 0.999 |
> |           | UPV            | 11.56 | 0.994 |
> |           | TS             | 14.9  | 0.996 |
> |           | KGW + D        | **13.48** | 0.994 |
> |           | Unigram + D    | **14.09** | 0.993 |
> |           | UPV + D        | **11.41** | 0.965 |
>
>
> | Model        | Method         | PPL   | AUC   |
> |----------|------------|----|-----|
> | Llama2 13B   | Wo Watermark   | 6.16  | -     |
> |              | KGW            | 12.30 | 0.997 |
> |              | Unigram        | 12.16 | 0.998 |
> |              | UPV            | 8.16  | 0.993 |
> |              | TS             | 7.62  | 0.998 |
> |              | KGW + D        | **7.57**  | 0.997 |
> |              | Unigram + D    | **7.58**  | 0.994 |
> |              | UPV + D        | **6.43** | 0.957 |
>
> > W2. In the comparison with MorphMark, the paper states that "MorphMark shows weaker detectability overall". But MorphMark also consistently shows better text quality, e.g., perplexity and acc in Table 1. Is it possible to tune the MorphMark model to achieve similar detectability, then compare text quality, or achieve similar text quality, then compare detectability?
>
> We appreciate your valuable comments. After extensively tuning the hyperparameters, we selected $k_{linear}$=9, which primarily controls the detection performance. We swept $k_{linear}$ in the range [1.5,9.0] with a step size of 0.5. Note that in the current paper, we used 1.55. We observed that our DynamicBias outperforms MorphMark. Furthermore, the token-level method of MorphMark suffers challenges in GSM datasets. We will update the draft to include the  following results and the detailed results from sweeping $k_{linear}$.
>
> ---
> C4 AVG
> | Method      | EN-PPL | EN-AUC  | KR-PPL | KR-AUC  | JA-PPL | JA-AUC  | AVG-PPL | AVG-AUC |
> |-------------|--------|---------|--------|---------|--------|---------|-------------|---------|
> | KGW         | 6.52   | 0.9912  | 6.34   | 0.9599  | 6.75   | 0.9708  | 6.53        | 0.9740  |
> | Unigram     | 6.45   | 0.9900  | 5.28   | 0.9668  | 6.18   | 0.9710  | 5.97        | 0.9759  |
> | UPV         | 6.11   | 0.9918  | 5.27   | 0.9370  | 5.66   | 0.9422  | 5.68        | 0.9570  |
> | SWEET       | 6.02   | 0.9893  | 5.79   | 0.9581  | 6.18   | 0.9664  | 6.00        | 0.9712  |
> | **Morpmark**    | 6.54   | 0.9895  | 7.32   | 0.9661  | 7.17   | 0.9637  | 7.01        | 0.9731  |
> | KGW + D   | **5.32**   | **0.9961**  | **5.43**   | **0.9825**  | **6.02**   | **0.9850**  | **5.59**        | **0.9878**  |
> | Unigram + D   | **5.23**   | **0.9934**  | **4.97**   | **0.9798**  | **5.75**   | **0.9783**  | **5.31**        | **0.9838**  |
> | UPV + D   | **5.23**   | **0.9953**  | **5.04**   | **0.9607**  | **5.48**   | **0.9604**  | **5.25**        | **0.9721**  |
>
> ---
> GSM AVG
> | Method      | EN-ACC | EN-AUC  | KR-ACC | KR-AUC  | JA-ACC | JA-AUC  | AVG-ACC | AVG-AUC |
> |-------------|--------|---------|--------|---------|--------|---------|---------|---------|
> | KGW         | 53.47  | 0.8481  | 8.19   | 0.8609  | 8.28   | 0.8845  | 23.31   | 0.8645  |
> | Unigram     | 52.47  | 0.8138  | 7.92   | 0.8568  | 8.02   | 0.8894  | 22.80   | 0.8533  |
> | UPV         | 52.12  | 0.7845  | 7.85   | 0.8078  | 7.71   | 0.8340  | 22.56   | 0.8088  |
> | SWEET       | 53.56  | 0.7717  | 8.47   | 0.8456  | 8.36   | 0.8632  | 23.46   | 0.8268  |
> | **Morpmark**    | 55.00  | 0.8220  | 8.38   | 0.8596  | 8.27   | 0.8843  | 23.88   | 0.8553  |
> | KGW + D   | 51.44  | **0.9661**  | 8.12   | **0.9494**  | **8.47**   | **0.9575**  | 22.68   | **0.9576**  |
> | Unigram + D   | 51.21  | **0.9461**  | **8.30**   | **0.9085**  | 7.77   | **0.9413**  | 22.43   | **0.9320**  |
> | UPV + D   | 47.86  | **0.9337**  | 7.58   | **0.8952**  | 7.43   | **0.9146**  | 20.96   | **0.9145**  |

---

> > ### Comment · Reviewer_5nZ9 · 2025-11-26
> > **Response to Rebuttal**
> >
> > I appreciate the authors for the additional experiments. I increased my score

---

> ### Author Response · Authors · 2025-11-27
>
> We appreciate your reply.

---

### Official Review · Reviewer_H6wU · 2025-11-11

**Soundness:** 2
**Presentation:** 3
**Contribution:** 2
**Rating:** 4
**Confidence:** 4

**Summary:**

This paper proposes DynamicBias, a sequence-aware calibration method for vocabulary-partition (green–red list) watermarking of large language models (LLMs). Instead of using a fixed bias $\delta$ for all tokens, DynamicBias computes a sequence-level average of logit-margin signals to scale the bias dynamically via a single parameter $\alpha$.
The authors show that this formulation admits a unique optimal $\alpha$ under a small-bias approximation, derive basic theoretical properties, and demonstrate improved detectability and robustness with competitive text quality across several LLMs  and datasets.

**Strengths:**

1.	Simple and Plug-and-Play Implementation: DynamicBias integrates easily into existing KGW-style watermarks without changing the hashing or detection pipeline, making it deployment-friendly.
2.	Comprehensive Evaluation: Experiments cover multiple models, languages, tasks, and paraphrasing attacks. Results consistently show moderate but reliable improvements in AUC with small quality loss.

**Weaknesses:**

1.	Limited Theoretical Novelty.
The idea of optimizing or calibrating the bias $\delta$ has been explored in prior works such as
Takezawa et al. (2023), Wouters (2024, ICML), and Cai et al. (2024), which already derive or analyze optimal biasing strategies for green–red list watermarking.
Compared to those, DynamicBias mainly introduces a sequence-level averaging heuristic and a light theoretical analysis under a small-bias approximation.
The claimed uniqueness of optimal α is mathematically straightforward and adds limited new insight.

2.	Approximate Theory.
The theoretical results rely on linearization and quadratic loss assumptions, without empirical verification of the underlying cost model. The "unique $\alpha$" result resembles a scaled-Lagrangian derivation rather than a new theoretical framework.

3.	Novelty Relative to Entropy-gated or Dynamic Bias Schemes.
DynamicBias is conceptually close to existing adaptive methods such as SWEET (entropy-gated) and MorphMark (token-level probability scaling). The main distinction—using sequence-averaged margin instead of entropy—may be incremental rather than groundbreaking.

4.	No Discussion of Computational Overhead or Detection Efficiency.
Since margin computation and averaging occur at every decoding step, runtime cost and practical detectability under large-scale inference scenarios are not quantified.


[1] Yuki Takezawa, Ryoma Sato, Han Bao, Kenta Niwa, and Makoto Yamada. Necessary and
sufficient watermark for large language models. arXiv preprint arXiv:2310.00833, 2023.

[2] Bram Wouters. Optimizing watermarks for large language models. In International Conference
on Machine Learning, pages 53251–53269. PMLR, 2024.

[3] Zhongze Cai, Shang Liu, Hanzhao Wang, Huaiyang Zhong, and Xiaocheng Li. Towards better
statistical understanding of watermarking llms. arXiv preprint arXiv:2403.13027, 2024.

**Questions:**

1.	How does DynamicBias perform relative to theoretically optimized schemes such as Takezawa et al. (2023) or Wouters (2024)? A small comparative or analytical discussion would strengthen the claim of novelty.
2.	The paper assumes a single global $\alpha$. Would learning or adapting α per domain/model further improve performance?
3.	How sensitive is the method to hyper-parameter c (top-k margin pool) in low-entropy or deterministic decoding settings?

---

> ### Author Response · Authors · 2025-11-21
>
> Hi! We sincerely appreciate your careful reading and feedback to improve our draft.
>
> > W1. Limited Theoretical Novelty. The idea of optimizing or calibrating the bias &delta; has been explored in prior works such as Takezawa et al. (2023), Wouters (2024, ICML), and Cai et al. (2024), which already derive or analyze optimal biasing strategies for green–red list watermarking. Compared to those, DynamicBias mainly introduces a sequence-level averaging heuristic and a light theoretical analysis under a small-bias approximation. The claimed uniqueness of optimal &alpha; is mathematically straightforward and adds limited new insight.
>
> We acknowledge that several prior works study how to optimize or calibrate the bias parameter in green/red watermarking. We will clarify more explicitly how our analysis differs from aforementioned papers.
> First, the existing theory focuses on token-level control of the bias as a function of the instantaneous distribution such as using green probability mass [1]. The most recent work, which is MorphMark, follows this line, and we already described the relation between such token-level methods and our sequence-level approaches in Appendix (line 912). In contrast, DynamicBias considers the running average of the logit margin between the top logit and ranks 2-5 and defines a single global scale α based on this sequence-level statistic.
>
> Our theoretical contribution is correspondingly sequence-level. Under a small-bias regime, we express the expected z-statistic of KGW-style detectors as a linear functional of the per-step green-mass gains and connect these gains to the sequence-averaged margin through Lemma 1 and Lemma 2, yielding an explicit gain-cost surrogate. While the calculus itself is simple, the novelty lies in identifying the appropriate sequence-level control variable and tying it directly to the KGW detection statistic. To the best of our knowledge, it is not covered by the cited works.
> Conceptually, prior adaptive methods operate on token-level entropy or probabilities, whereas DynamicBias uses the sequence-averaged logit margin as its control signal and applies a single global parameter α, which we analyze under a small-bias regime and find to work consistently across different models, domains, and languages. While we do not claim this to be groundbreaking, we believe that the  model-, domain-, and language-agnostic nature of DynamicBias provides a meaningful and practically relevant contribution beyond existing adaptive watermarking schemes.
>
> We will update the draft by explicitly positing our analysis as a sequence-level control law for KGW-style watermarking, rather than a general replacement of existing theory by moving the relevant explanations from the Appendix into the main body. We will also expand the related work discussion to incorporate the suggested references.
>
> [1] MorphMark: Flexible Adaptive Watermarking for Large Language Models, ACL2025
>
>
> > W2. Approximate Theory. The theoretical results rely on linearization and quadratic loss assumptions, without empirical verification of the underlying cost model. The "unique α" result resembles a scaled-Lagrangian derivation rather than a new theoretical framework.
>
> We acknowledge that our analysis is approximate and relies on a small-bias regime with linearized gain and quadratic cost. Our intent is not to claim an entirely new theoretical framework, but to provide a principled local justification for why a single global α on the sequence-averaged margin is sufficient in practice.
>
> Regarding empirical verification of the cost model, Sec 4.2 and Sec 4.6 already provide empirical support for the gain-cost view underlying our small-bias analysis. Sec 4.2 shows a strong negative correlation (Pearson r=-0.960) between the sequence-averaged logit gap and AUC across models, domains, and languages, indicating that the gain from a fixed static bias depends strongly on the margin statistics of the next-token distribution. Sec 4.6 then directly ablates the scaling factor α and shows that the KGW z-score grows almost linearly with α, while quality metrics (PPL and ACC) exhibit mild quadratic curvature with very high $R^2$ values. This behavior matches the linear-gain / quadratic-cost assumptions used in Lemma 1, Lemma 2, and Theorem 1.

---

> ### Author Response · Authors · 2025-11-21
>
> > W3. Novelty Relative to Entropy-gated or Dynamic Bias Schemes. DynamicBias is conceptually close to existing adaptive methods such as SWEET (entropy-gated) and MorphMark (token-level probability scaling). The main distinction—using sequence-averaged margin instead of entropy—may be incremental rather than groundbreaking.
>
>
> We appreciate the opportunity to clarify the differences. A key practical diction is at detection time. SWEET requires explicitly querying the model to obtain token-level probabilities (or entropies) for each generated sequence, which incurs substantial computational overhead during detection. In contrast, DynamicBias is designed to remain fully compatible with the standard KGW detector and does not require any additional model calls or entropy computation at detection time.
>
> Conceptually, prior adaptive methods operate on token-level entropy or probabilities, whereas DynamicBias uses the sequence-averaged logit margin as its control signal and applies a single global parameter α, which we analyze under a small-bias regime and find to work consistently across different models, domains, and languages. While we do not claim this to be groundbreaking, we believe that the  model-, domain-, and language-agnostic nature of DynamicBias provides a meaningful and practically relevant contribution beyond existing adaptive watermarking schemes.
>
> > W4. No Discussion of Computational Overhead or Detection Efficiency. Since margin computation and averaging occur at every decoding step, runtime cost and practical detectability under large-scale inference scenarios are not quantified.
>
> We appreciate your careful reading and constructive feedback. Following your feedback, we conducted additional experiments using three models of different sizes. To evaluate runtime, we sampled 100 instances from the C4 dataset, generated 200 tokens for each instance, and computed the average latency on an NVIDIA A6000 GPU. We also discussed computational overhead in Appendix A.4.
>
> The results show that Dynamic Bias does not significantly increase generation time compared to the baseline methods. At detection time, DynamicBias incurs almost no additional overhead because it does not require computing token entropies or querying the model, unlike some entropy-based methods. We will incorporate these results and a detailed description of the setup into the revised draft.
>
> | Model          | Method     | Generation Time (token/ms) | Generation Time (data/s) | Detection Time (data/ms) |
> |----------------|------------|-----------------------------|----------------------------|----------------------------|
> | **Llama 3.2 1B** | KGW        | 11.75 | 2.30 | 25.1 |
> |                | KGW + D   | 11.90 | 2.31 | 25.0 |
> |                | UPV        | 14.23 | 2.78 | 41.5 |
> |                | UPV + D   | 14.38 | 2.78 | 41.5 |
> |                | SWEET      | 11.88 | 2.30 | 68.4 |
> |                | Morphmark  | 12.14 | 2.33 | 26.1 |
> |----------------|------------|-----------------------------|----------------------------|----------------------------|
> | **Llama 3.2 3B** | KGW        | 26.28 | 5.25 | 24.9 |
> |                | KGW + D   | 26.41 | 5.28 | 25.0 |
> |                | UPV        | 28.61 | 5.71 | 38.5 |
> |                | UPV + D   | 28.74 | 5.74 | 38.9 |
> |                | SWEET      | 26.37 | 5.27 | 133.6 |
> |                | Morphmark  | 26.64 | 5.28 | 26.8 |
> |----------------|------------|-----------------------------|----------------------------|----------------------------|
> | **Qwen 3 4B**  | KGW        | 34.93 | 6.97 | 26.2 |
> |                | KGW + D   | 35.17 | 7.03 | 26.3 |
> |                | UPV        | 37.92 | 7.57 | 44.1 |
> |                | UPV + D   | 38.03 | 7.61 | 44.6 |
> |                | SWEET      | 35.18 | 7.01 | 163.4 |
> |                | Morphmark  | 35.54 | 7.06 | 28.1 |

---

> ### Author Response · Authors · 2025-11-21
>
> > Q1. How does DynamicBias perform relative to theoretically optimized schemes such as Takezawa et al. (2023) or Wouters (2024)? A small comparative or analytical discussion would strengthen the claim of novelty.
>
> Thank you for your thoughtful advice. We address this point in our response to Weakness 1.
>
> > Q2. The paper assumes a single global alpha. Would learning or adapting α per domain/model further improve performance?
>
> Thank you for your thoughtful advice. We agree that, in principle, learning alpha per domain or per model could further improve the quality-detectability trade-off, as we also discussed in the final sentence of the conclusion in the draft.
>
> > Q3. How sensitive is the method to hyper-parameter c (top-k margin pool) in low-entropy or deterministic decoding settings?
>
> We would first like to clarify a potential misunderstanding. In this paper, all experiments except those in Tables 11 and 12 were conducted under deterministic decoding settings. Therefore, the analysis of the hyper-parameter c in Table 4 also corresponds to deterministic decoding.
>
> Following your suggestion, we conducted additional experiments to analyze the sensitivity under low-entropy settings on the HumanEval dataset. The results show that, as c increases, AUC increases, while pass@1 decreases moderately, which is consistent with the trade-off pattern observed in Table4.
> We will include these results.
>
> | c  | Pass@1  | AUROC  |
> |----|---------|--------|
> | 2  | **0.3171**  | 0.7382 |
> | 5  | 0.3049  | 0.8178 |
> | 10 | 0.2866  | **0.8530** |

---

### Author Response · Authors · 2025-11-23
**Dear reviewrs**

> For the novelty of our work

We  would  like  to  clarify  the  novelty  of  our  draft.  Our  draft  does  not  simply  use  sequence-aware  DynamicBias  but  also  proposes  a  model-,  domain-,  and  language-independent  approach for the trade-off between the text quality and detection performance.  This  has  not  been  discussed  before,  and  to  support  this,  we  provide  theoretical  analysis  and  empirical  results  that  strongly  support  our  theorem.  We  believe  this  is  not  a  limited  novelty,  but  rather  a  step  in  the  right  direction  for  real-world  applicability  of  LLM  watermarking.

---

### Author Response · Authors · 2025-11-25

We thank the reviewers for their insightful and constructive feedback. We have revised the draft to reflect this feedback as much as possible.

---

### Author Response · Authors · 2025-11-29
**Summary of Our Contributions and Responses to Weakness**

## **Summary of Our Contributions**

- DynamicBias considers sequence-aware calibration for the trade-off between text quality and detection performance. This has not been discussed before, and to support this, we provide theoretical analysis and empirical results that strongly support our theorem.

- Different from previous token-level approaches for adaptive bias, DynamicBiasis  a model-, domain-, and language-independent approach for the trade-off between text quality and detection performance. We believe this is not a limited novelty, but rather a step in the right direction for real-world applicability of LLM watermarking.

- Our experimental results strongly support our theorem (Figure 3). Furthermore, experimental results and in-depth analysis show that token-level adaptive bias methods suffer from instability due to fluctuations (blue curve in Figure 5).

- We empirically show that our method can be adapted to any KGW-based methods with improving performance across multiple languages (English, Korean, Japanese) on open-ended generation tasks (C4 and mC4) and mathematical reasoning tasks (GSM and MGSM). We also conducted document summarization and code generation tasks to show its generalizability.


## **Responses to Weakness Pointed out by Reviewers**

## **Reviewer H6wU:**

> Limited Theoretical Novelty and Approximate Theory:

**We would like to emphasize that we have already discussed the relation with the token-level theorem.** In the revised draft, we moved this discussion into the main body (Section 3.4). Furthermore, **we have already shown the linearization and quadratic loss assumption with empirical verification as Reviewer jV92 and 5nZ9 recognized (Figure 3)**. In addition, unlike previous token-level approaches, our proposed method targets a model-, domain-, and language-independent (As reviewer 5nZ9 noted, MorphMark showed weaker detectability and required extensive parameter tuning for a single domain). Thus, our straightforward theorem is intended to provide a clean and general formulation that captures the essential behavior of DynamicBias without relying on model-specific assumptions, which we believe is appropriate for a practical and widely applicable watermarking framework.

> Novelty Relative to Entropy-gated or Dynamic Bias Schemes:

**We would like to clarify potential misunderstanding for SWEET (entropy-gated).** SWEET requires explicitly querying the model to obtain token-level probabilities (or entropies) for each generated sequence, which incurs substantial computational overhead during detection (Table 7).

No Discussion of Computational Overhead or Detection Efficiency:

**We would like to clarify a potential misunderstanding. We have already discussed computation overhead as Reviewer jV92 recognized (Appendix A.4).** We also provided the computational comparison in the revised draft (Table 7). DynamicBias outperforms other methods when integrated with KGW, in both generation time and detection time, while incurring minimal overhead.

**Reviewer 5nZ9 (improved scores to 6):**

> Comparison to a model-dependent method for learning bias:

We conducted additional experiments and ours outperforms even a model-dependent method for adaptive method for bias. We included this in the revised draft in Table 11.

> MorphMark parameter tuning:

We extensively tuned MorphMark and showed the limited generalizability due to token-level adaptive approach (Table 6, 13 and Figure 6).

**Reviewer jV92:**

> Report generation throughput or wall-clock time relative to vanilla watermarking. Even if the per-step overhead is small, empirical confirmation would strengthen claims of practical deployability:

We also provided the computational comparison in the revised draft (Table 7). DynamicBias outperforms other methods when integrated with KGW, in both generation time and detection time, while incurring minimal overhead.


> While results are strong overall, the paper could better quantify trade-offs—e.g., where DynamicBias might underperform in high-entropy tasks or non-English domains (given these were tested):

We revised the draft by discussing it (lines 325-327).

**Reviewer L388:**

> The method shows limited conceptual novelty, as it is quite similar to Morphmark, which also employed a token-level dynamic bias:

Please refer to summary of our contribution.

---

### Meta-Review · Area_Chair_b5PE · 2026-01-07

**Summary:**

The paper introduces DynamicBias, a new way to watermark text generated by large language models so we can reliably tell whether a model wrote it. Instead of using one fixed watermark strength for every token, it adjusts the strength on the fly based on how confident the model is at each step. This makes the watermark easier to detect, more consistent across different models and languages, and harder to remove through paraphrasing, all while keeping the text quality high.

Multiple reviewers (H6wU, L388, jV92) converged on the assessment that the paper proposes an incremental extension to existing
adaptive watermarking methods. While the empirical results are generally solid and the method is carefully implemented, a central
concern is that the conceptual distinction between DynamicBias and existing token-level adaptive biasing approaches remains
insufficiently articulated.

In particular, although the authors repeatedly emphasize the advantages of sequence-level averaging, they do not provide a clear
analysis of the limitations of token-level biasing itself. This is especially notable given that DynamicBias still applies perturbations to
the logits at every token during generation. As a result, it is not entirely clear what fundamentally differentiates DynamicBias from
token-level biasing beyond smoothing the control signal, nor why token-level biasing should be viewed as suboptimal in principle
rather than merely noisier in practice.

Additionally, the paper does not provide a sufficiently explicit discussion of the implications of using a single global scaling
parameter α. Several reviewers questioned whether a single global α can be appropriate across models, domains, and entropy
regimes, and this concern remains only partially addressed.

**Reviewer Concerns:**

Concerns addressed by the rebuttal:
+ Computational overhead and deployability: Reviewers H6wU and jV92 raised concerns regarding runtime cost and detection efficiency. The authors addressed this well by providing detailed generation and detection time experiments across multiple model sizes, demonstrating minimal overhead.
+ Missing baselines: Following Reviewer 5nZ9’s request, the authors added comparisons with additional adaptive baselines, including model-dependent methods, which strengthened the empirical evaluation.
+ Trade-offs and limitations: Reviewers jV92 and L388 noted that the original submission insufficiently discussed quality–detectability trade-offs and limitations. The authors responded with additional ablation studies and hyperparameter analyses, which substantially improved clarity.
+ Performance on very short sequences: Reviewers jV92 and L388 questioned whether sequence-level averaging might weaken watermarking on short texts. The authors addressed this concern with targeted experiments analyzing detectability as a function of sequence length.

Concerns still outstanding:
- Limited novelty: All reviewers (H6wU, L388, jV92) questioned the novelty of the contribution relative to prior adaptive watermarking methods. While the rebuttal clarified positioning, it did not fundamentally change this assessment.
- Single global \alpha: Reviewer H6wU explicitly questioned the implications of using a single global scaling parameter. The rebuttal acknowledged this limitation but did not provide a concrete analysis or resolution.
- Comparison with MorphMark: Reviewer 5nZ9 noted that MorphMark may achieve comparable performance with sufficient hyperparameter tuning. While additional experiments were provided, the discussion of MorphMark's limitations remains largely empirical and does not clearly articulate why token-level adaptive methods are intrinsically disadvantaged.
- Token-level vs. sequence-level distinction: Reviewer L388 raised questions about the effect of token-level certainty signals \bar{m}_t on watermarking behavior. The authors
did not provide a principled discussion or targeted experiments that directly analyze why token-level biasing is undesirable, leaving this conceptual distinction insufficiently
resolved.

**Reviewer Scores:**

+ Reviewer H6wU: Likely to maintain their original score. The rebuttal addressed practical concerns but did not resolve the reviewer’s core reservations regarding novelty and theoretical depth.
+ Reviewer 5nZ9: Likely to maintain their score. While additional baselines were added, concerns regarding MorphMark and conceptual differentiation persist.
+ Reviewer jV92: Likely to maintain or slightly lower their score. Although many practical concerns were addressed, unresolved conceptual issues may limit enthusiasm.
+ Reviewer L388: Likely to maintain their score. The rebuttal improved empirical coverage but did not fully address questions about token-level versus sequence-level biasing.

---

### Decision · Program_Chairs · 2026-01-26

Reject